# Identification of pharmacological inducers of a reversible hypometabolic state for whole organ preservation

Megan M Sperry[1,2], Berenice Charrez[1], Haleh Fotowat[1], Erica Gardner[1], Kanoelani Pilobello[1], Zohreh Izadifar[1], Tiffany Lin[1], Abigail Kuelker[1], Sahith Kaki[1], Michael Lewandowski[1], Shanda Lightbown[1], Ramses Martinez[1], Susan Marquez[1], Joel Moore[1], Maria Plaza-Oliver[1,3], Adama M Sesay[1], Kostyantyn Shcherbina[1], Katherine Sheehan[1], Takako Takeda[1], Daniela Del Campo[1], Kristina Andrijauskaite[4], Exal Cisneros[4], Riley Lopez[4], Isabella Cano[4], Zachary Maxwell[4], Israel Jessop[4], Rafa Veraza[4], Leon Bunegin[4], Thomas J Percival[5], Jaclyn Yracheta[5], Jorge J Pena[5], Diandra M Wood[5], Zachary T Homas[5], Cody J Hinshaw[5], Jennifer Cox-Hinshaw[5], Olivia G Parry[5], Justin J Sleeter[5], Erik K Weitzel[5], Michael Levin[1,2,6], Michael Super[1], Richard Novak[1], Donald E Ingber[7,8]*

[1]Wyss Institute for Biologically Inspired Engineering at Harvard University, Boston, United States; [2]Department of Biology, Tufts University, Medford, United States; [3]DEVANA group, Faculty of Pharmacy, University of Castilla-La Mancha, Ciudad Real, Spain; [4]Vascular Perfusion Solutions Inc, San Antonio, United States; [5]RESTOR, 59th Medical Wing, JBSA, Lackland AFB, San Antonio, United States; [6]Allen Center, Tufts University, Medford, United States; [7]Vascular Biology Program & Department of Surgery, Boston Children's Hospital and Harvard Medical School, Boston, United States; [8]Harvard John A. Paulson School of Engineering and Applied Sciences, Boston, United States

*For correspondence:
don.ingber@wyss.harvard.edu

**Abstract** Drugs that induce reversible slowing of metabolic and physiological processes would have great value for organ preservation, especially for organs with high susceptibility to hypoxia-reperfusion injury, such as the heart. Using whole-organism screening of metabolism, mobility, and development in *Xenopus*, we identified an existing drug, SNC80, that rapidly and reversibly slows biochemical and metabolic activities while preserving cell and tissue viability. Although SNC80 was developed as a delta opioid receptor activator, we discovered that its ability to slow metabolism is independent of its opioid modulating activity as a novel SNC80 analog (WB3) with almost 1000 times less delta opioid receptor binding activity is equally active. Metabolic suppression was also achieved using SNC80 in microfluidic human organs-on-chips, as well as in explanted whole porcine hearts and limbs, demonstrating the cross-species relevance of this approach and potential clinical relevance for surgical transplantation. Pharmacological induction of physiological slowing in combination with organ perfusion transport systems may offer a new therapeutic approach for tissue and organ preservation for transplantation, trauma management, and enhancing patient survival in remote and low-resource locations.

## eLife assessment

Pharmacological induction of physiological slowing combined with organ perfusion systems could provide a novel therapeutic strategy for tissue and organ preservation. Using a *Xenopus* model, the

authors provide **important** findings on a use of drug to slow down metabolism for the purpose of organ preservation. The authors provide **compelling** evidence that SNC80 can rapidly and reversibly slow biochemical and metabolic activities while preserving cell and tissue viability. This approach may be beneficial for transplantation, trauma management, and improving organ survival in remote and low-resource settings

## Introduction

Tissue and organ loss to trauma, disease, and physical injury account for a large proportion of human ailments and approximately $400 billion in annual medical burden (*Vela et al., 2020*). The rapid and reversible slowing of metabolic and other physiological processes, here referred to as 'biostasis', can improve the survival of cells and organs for transplantation. This is currently accomplished clinically by lowering temperature and static cold storage is the standard of care for organ and tissue preservation; however, its long-term use can cause damage to the integrity of the graft (*Vela et al., 2020*; *Petrenko et al., 2019*; *Guibert et al., 2011*). Hypothermia has also been used to induce a state of biostasis clinically using ex vivo machine perfusion technologies, e.g., during cardiac transplant surgery. Combination of protective agents with perfusion and/or partial freezing approaches have extended preservation times in rat livers and whole pigs (*Andrijevic et al., 2022*; *Tessier et al., 2022*), and neural modulation approaches have been successful at demonstrating central control of body temperature and general metabolic state (*Hrvatin et al., 2020*; *Takahashi et al., 2020*). However, both mechanical cooling and neural stimulation approaches are challenging to implement in a trauma triage or resource-limited situations, which would be better addressed by pharmaceutical interventions.

Several molecular strategies have been proposed to achieve metabolic suppression for organ preservation, including modulating $H_2S$, AMPK, opioid receptors, microRNAs, HO-1, and Nrf2 (*Dou et al., 2022*). $H_2S$ exposure has been reported to induce a hypometabolic state (reduced metabolism and body temperature) in a rodent model, although this was later found to be mediated in part by a low baseline laboratory temperature (*Blackstone et al., 1979*). The synthetic opioid peptide DADLE can also induce hypometabolism and improve organ preservation via a delta opioid receptor (DOR) mechanism, but with varied results (*Beal et al., 2019*; *Ratigan and McKay, 2016*). Compounds that directly impact glycolysis and mitochondrial respiration also have been considered for modulating organ metabolism, but many are acutely toxic and not reversible, fail to protect tissues from ischemic injury, or lack drug-like properties. Specifically, reduced functionality of the mitochondrial electron transport chain system without the endogenous antioxidant and anti-inflammatory molecules released by hibernators can result in the release of reactive oxygen species and cause tissue damage (*Dou et al., 2022*).

Despite these efforts, there remains a large unmet need for improved tissue and organ preservation approaches across multiple application areas. Thus, in the present study, we set out to identify small molecules that could slow metabolism and mimic states normally induced by hypothermia, hibernation, or torpor, which could be used for preservation of living cells, tissues, and organs ex vivo, and potentially in vivo as well. Specifically, we required this biostasis approach to be rapidly inducible (<1 hr) and safely reversible, with major tissue functions returning to control levels within 24 hr.

Through survey of the literature, we identified drugs with unintended side effects, such as lowering body temperature, which suggested they might have broad effects on metabolism. During early compound screening, we evaluated known cellular respiration inhibitors (such as rotenone, FCCP, oligomycin A, and Ru360) and torpor inducers from the literature (such as DADLE, JS-K, AMP, sodium sulfide nonahydrate, and BW373u86). Compounds involved in activating the DOR pathway were of particular interest based on prior work suggesting that DOR activation extends the hypothermic preservation time of organs, including the heart (*Schultz et al., 1997*). Among these drugs was SNC80, a small molecule drug originally developed as a nonaddictive pain reliever that acts independently of the mu-opioid pathway (*Rawls et al., 2005*). This drug caught our attention because it has been shown to induce hypothermia and protect against the effects of spinal cord ischemia in rodents (*Rawls et al., 2005*; *Horiuchi et al., 2004*), however, further study was required to understand if these observations were related to biostatic effects. To explore whether SNC80 had potential to broadly slow tissue metabolism and physiology, we administered the drug to *Xenopus laevis* embryos and tadpoles, which we and others have previously shown to be useful models for drug screening due to

their small size, extrauterine development, and skin permeability to small molecules (*Sperry et al., 2022*; *Novak et al., 2022*; *Schmitt et al., 2014*; *Sullivan and Levin, 2018*). The *Xenopus* tadpoles were dosed at higher levels than dictated by SNC80's defined IC50 value, with the goal of exploring the effects of SNC80 outside its known pain relief properties and accentuating the drug's 'off-target' effects, including hypometabolism.

## Results

### Demonstration of physiological slowing in *Xenopus*

We first assessed SNC80's impacts on mobility, metabolism, and heart activity in *Xenopus*. Behavioral assessment in mobile tadpoles showed that 100 μM SNC80 slows movement relative to vehicle-treated counterparts, reducing activity by approximately 50% within 1 hr, which is rapidly reversible when the drug is removed (*Figure 1a*). SNC80 also suppressed the rate of oxygen consumption to one-third of baseline within 3 hr of treatment (*Figure 1b*) and had extreme suppressive, but fully reversible, effects on heart rate after only 1 hr of exposure (*Figure 1c*). Oxygen consumption was reduced in immobile *Xenopus* embryos treated with SNC80 as well (*Figure 1—figure supplement 1*), suggesting that its metabolism-suppressing effects are independent of the slowed movement observed in mobile tadpoles.

Using in situ matrix-assisted-laser desorption/ionization-time of flight mass spectrometry imaging (MALDI-ToF MSI), we assessed and visualized the biodistribution of SNC80 in the *Xenopus* tadpole. SNC80 was detected within 1 hr of treatment and it appeared in a punctate distribution in the gastrointestinal tract, gill region, and skeletal muscle (*Figure 2a and b*), suggesting full-body delivery of the drug. Specifically, uptake observed in the muscle may be responsible for the slowed motion observed in *Xenopus* activity assays. Past lipidomic studies of hibernating mammals demonstrated the importance of lipid molecules for thermal adaptation during hibernation with significant alterations of lipidomic profiles being observed in liver, plasma, brain, skeletal muscle, and cardiac muscle during torpor as compared to the animal's active state (*Giroud et al., 2019*; *Kolomiytseva, 2011*). Similarly, MALDI-ToF MSI analysis revealed significantly higher levels of acylcarnitine and cholesterol ester in SNC80-treated groups in both skeletal muscle and brain, but not in cardiac tissue (*Figure 2c*). Increased levels of long chain acylcarnitine associated with enhanced mitochondrial fatty acid oxidation are observed during fasting as well as in hibernating brown bears (*Soeters et al., 2009*; *Welinder et al., 2016*), and cholesterol ester levels increase in hibernating ground squirrels (*Otis et al., 2011*). While the increased acylcarnitine and cholesterol ester levels in brain suggest a shift toward beta oxidation of fatty acids, they also have been implicated in other pathways including antioxidant activity and neurotransmission (*Jones et al., 2010*; *Petrov et al., 2016*).

Having confirmed that SNC80 induces slowing of multiple physiological parameters in the whole *Xenopus* organism, we then set out to explore if the molecular basis for these effects is dependent on the DOR pathway. This was important since pre-clinical testing of SNC80 in rodents and non-human primates has demonstrated seizure side effects at higher doses, possibly due to SNC80's direct inhibition of forebrain GABAergic neurons via DOR activity (*Chu Sin Chung et al., 2015*). As SNC80 is a known DOR agonist (*Rawls et al., 2005*; *Horiuchi et al., 2004*), we next evaluated whether its effects could be mediated by delta opioids. However, when we tested the DOR antagonist, naltrindole (NTI), at double the concentration (200 μM) relative to SNC80 (100 μM), it did not block the hypometabolic effects of SNC80 (*Figure 2d*), suggesting that SNC80 modulates metabolism independent of DOR activation.

### Attenuating the delta opioid activity of SNC80

To further test if the physiological slowing effects of SNC80 are independent of DOR activation, we used a medicinal chemistry approach to explore whether we could separate its metabolism slowing and DOR activities. Toward this end, we designed molecular entities void of the distal basic nitrogen (N-ally group within SNC80), which we hypothesized was a critical DOR pharmacophore. This effort led to the development of a novel morpholino compound, WB3 (*Figure 3a* and Supplemental Synthesis Description), which was synthesized and subsequently tested for DOR binding activity using a radioligand binding assay and nonlinear fitting for specific binding to calculate the Hill slope and IC50 value. A specific binding curve could not be fit for SNC80 due to its high affinity and therefore its

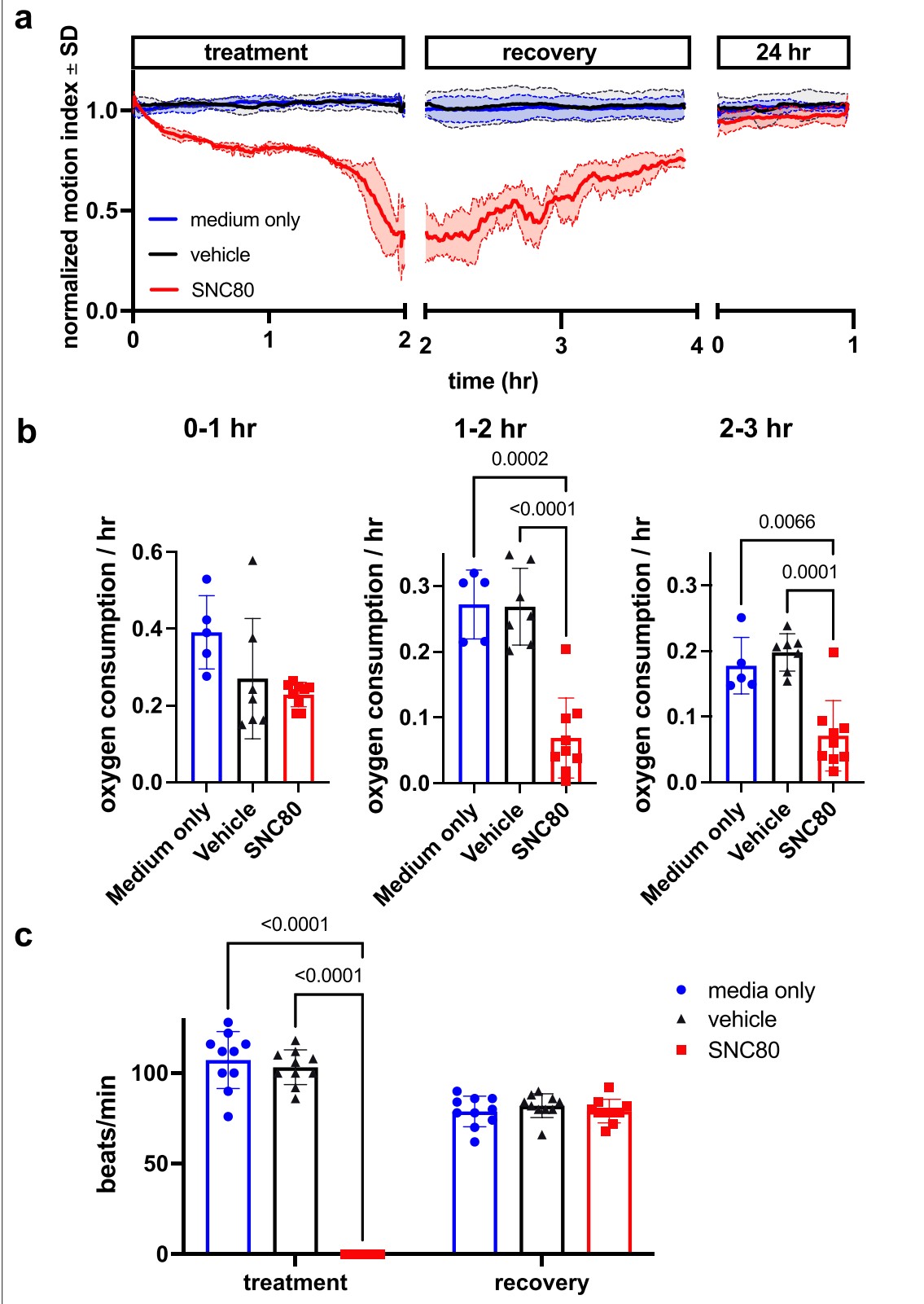

**Figure 1.** SNC80 treatment in *Xenopus* tadpoles. (**a**) Swimming activity of *Xenopus* during 100 µM SNC80 or vehicle treatment. Data represent the mean ± SD of n=3 replicates, with 10 tadpoles per replicate. (**b**) Oxygen consumption rate for *Xenopus* treated with SNC80 or vehicle controls. N=5 medium only, n=7 vehicle and n=9 SNC80, with each data point representing the cumulative oxygen consumption for 5 tadpoles. (**c**) Heart rate in SNC80 and vehicle-treated *Xenopus* (n=10 tadpoles/group). Statistical comparisons were performed using a Brown-Forsythe and Welch's ANOVA test

*Figure 1 continued on next page*

*Figure 1 continued*

with a Dunnett correction for multiple comparisons (**b**) and a two-way ANOVA (treatment × timepoint) with Tukey correction for multiple comparisons (**c**). Bar plots show the mean ± SD of each group.

The online version of this article includes the following figure supplement(s) for figure 1:

**Figure supplement 1.** Metabolic measurements in SNC80-treated *Xenopus* embryos.

IC50 is <4.6 nM, whereas WB3 exhibited an IC50 of 3470 nM (*Figure 3b*), suggesting almost 1000 times less DOR binding activity for WB3 compared to SNC80. Subsequently, we screened WB3 in the *Xenopus* swimming, oxygen consumption, and heart rate assays and found that the analog retains physiological slowing properties in vivo despite lacking delta opioid activity (*Figure 3c, d, and e*). In fact, we observed more potent and rapid effects on swimming with exposure to WB3 compared to SNC80, suggesting that the non-opioid properties of SNC80 may be primarily responsible for its observed physiological effects.

## Enhanced preservation of whole pig hearts and limbs perfused ex vivo

Based on these intriguing findings with *Xenopus* tadpoles, we set out to explore the potential clinical value of SNC80. Cardiac transplant is the gold standard for treating end-stage heart failure; however, it is hindered by donor shortage, limited cardiac graft preservation time, and lack of optimal organ preservation conditions. Currently, hearts are allocated to patients located within 4 hr of organ removal from the donor. Thus, to explore the potential utility of biostasis induction to lengthen the time between organ donation and recipient surgery through pharmacological induction of a hypometabolic state, we perfused surgically explanted whole porcine hearts with SNC80 using a portable, oxygenated perfusion preservation device (VP.S ENCORE) at subnormothermic temperatures (20–23°C). Prior work has demonstrated that the VP.S ENCORE enhances cardiac viability over standard of care-static cold storage and hearts perfused in this device for 4 hr demonstrate similar cardiac function to a healthy heart in both preclinical animal models and human deceased donor hearts (*Veraza et al., 2023*; *Andrijauskaite et al., 2023*).

We investigated the application of a biostasis inducer for preservation of the heart during 6 hr of perfusion in the VP.S ENCORE device. Importantly, with SNC80 treatment, we observed a rapid decline in the heart's oxygen consumption to <50% of that observed in hearts exposed to the vehicle control, which was sustained over the 6 hr preservation period (*Figure 4a*). After preservation, hearts were removed from the VP.S ENCORE and placed in a Langendorff system for reperfusion and evaluation of oxygen consumption and cardiac contractility. This system assesses the left ventricular dP/dT without preload and afterload, thereby allowing the evaluation of the intrinsic contractility of the left ventricle (LV). After 30 min of reperfusion and before defibrillation, SNC80-treated hearts had a lower oxygen consumption level than vehicle-treated hearts (2.39 ± 1.23 versus 4.28 ± 2.48 mL/min/100 g, respectively, at the 30 min timepoint) (*Figure 4b*) that recovered to normal levels after defibrillation and with increases in temperature (2.49 ± 0.53 and 2.35 ± 0.31 mL/min/100 g, respectively, at 60 min post defibrillation) (*Figure 4c*). No evidence of substantial change in LV contractility or relaxation were detected in SNC80-exposed hearts, either before or after they were defibrillated and exposed to epinephrine (*Figure 4d*).

Gene expression analysis of heart biopsy samples showed significant reduction of markers of inflammation (IL-6, IL-8, TNFα), hypoxia (Hif1α), and cell death (TP53, BCL2) in SNC80-treated hearts when compared to vehicle control hearts (*Figure 4e*), and no changes in the mitochondrial ATP synthase marker, ATP5MC3, were observed. Histopathological evaluation showed retention of normal muscle morphology (*Figure 4f*) without signs of tissue edema immediately after perfusion in the VP.S ENCORE. In some hearts treated with SNC80 greater waviness of muscle fibers was observed, possibly indicating a state of muscle contraction.

Electrocardiogram (ECG) data revealed that both vehicle and SNC80-treated hearts exhibited irregular beating after perfusion and prior to defibrillation. After defibrillation by electric shock and epinephrine delivery, the P and QRS waveforms were visible in ECGs from three of four SNC80-treated hearts (*Supplementary file 1*), indicating that those hearts regained atrial and ventricular depolarization (internal charge of cardiac cells become less negative). The presence of the P and QRS waveforms suggests that the intrinsic atrial and ventricular muscle fiber contractility was preserved

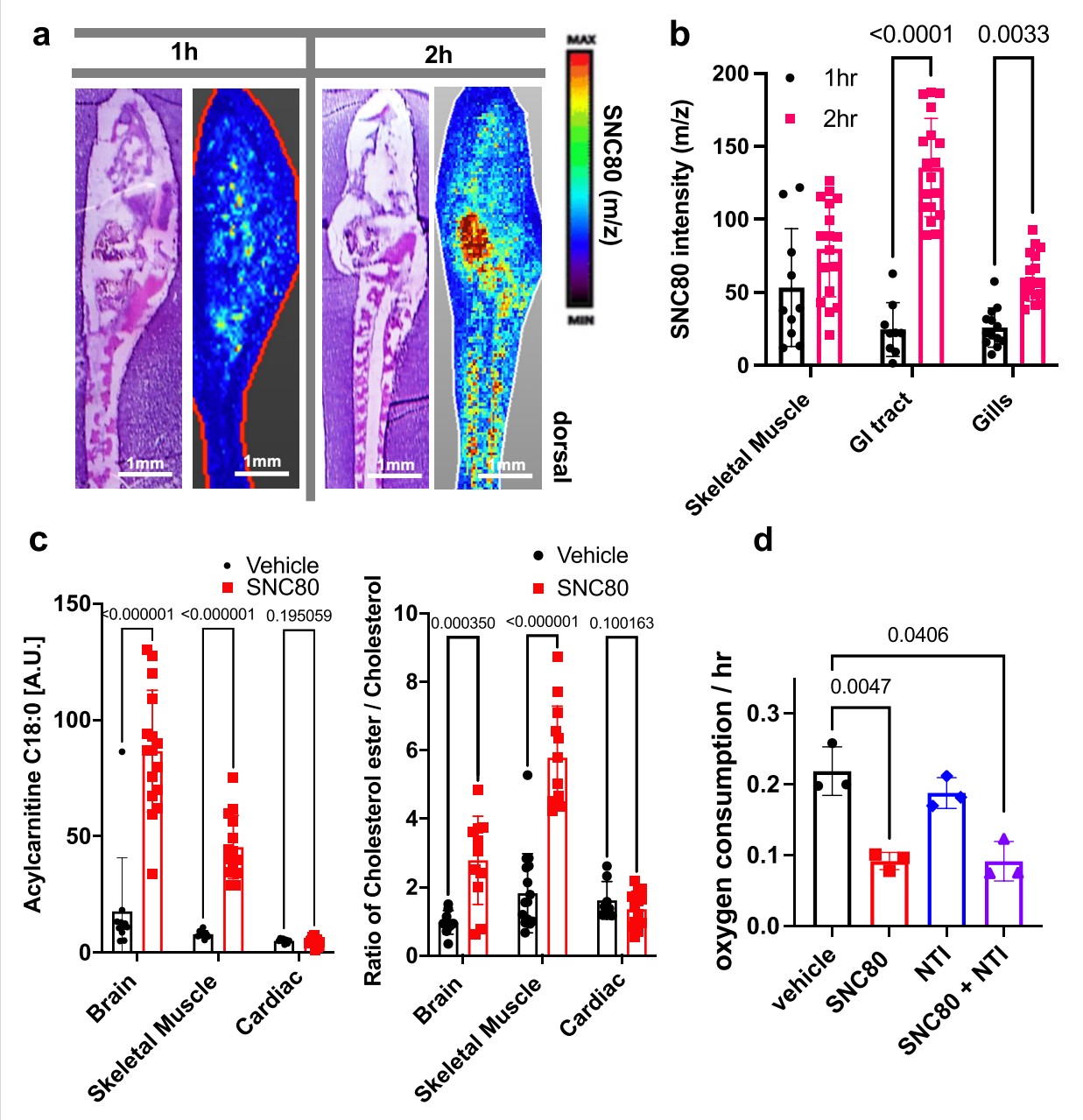

**Figure 2.** SNC80 uptake and activity at the delta opioid receptor. (**a**) Distribution of SNC80 in *Xenopus* tadpoles after 1 and 2 hr of compound exposure. (**b**) Uptake of SNC80 in the tadpole skeletal muscle, gastrointestinal (GI) tract, and gills at 1 and 2 hr of exposure. (**c**) Levels of acylcarnitine and cholesterol ester in the skeletal muscle, brain, and cardiac tissue after 1 hr of SNC80 treatment. In vivo distributions of SNC80 and lipid levels measured from N=5 tadpoles per condition; N=3 sections/slide. (**d**) Oxygen consumption in *Xenopus* tadpoles treated with SNC80, the delta opioid antagonist naltrindole, or a combination of SNC80 and naltrindole. N=3 replicates/group with each data point representing the cumulative oxygen consumption from 5 tadpoles. Statistical comparisons were performed using a two-way ANOVA (time × tissue region) with Sidak's correction for multiple comparisons (**b**), multiple unpaired t-tests for each tissue region with FDR correction (**c**), and a Welch's ANOVA test with Dunnett correction for multiple comparisons between vehicle and each treatment group (**d**). Bar plots show the mean ± SD of each group.

and the overall conduction system of the heart was viable. The pulse rates of SNC80-treated hearts were at or near normal range for porcine hearts (70–120 beats/min) after defibrillation. Vehicle-treated hearts exhibited tachycardia following defibrillation, with all exhibiting pulse rates above the normal range for porcine hearts.

We also tested organ preservation in porcine limbs, where prolonged ischemia after amputation rapidly decreases the viability of skeletal muscle. Current preservation methods limit the success

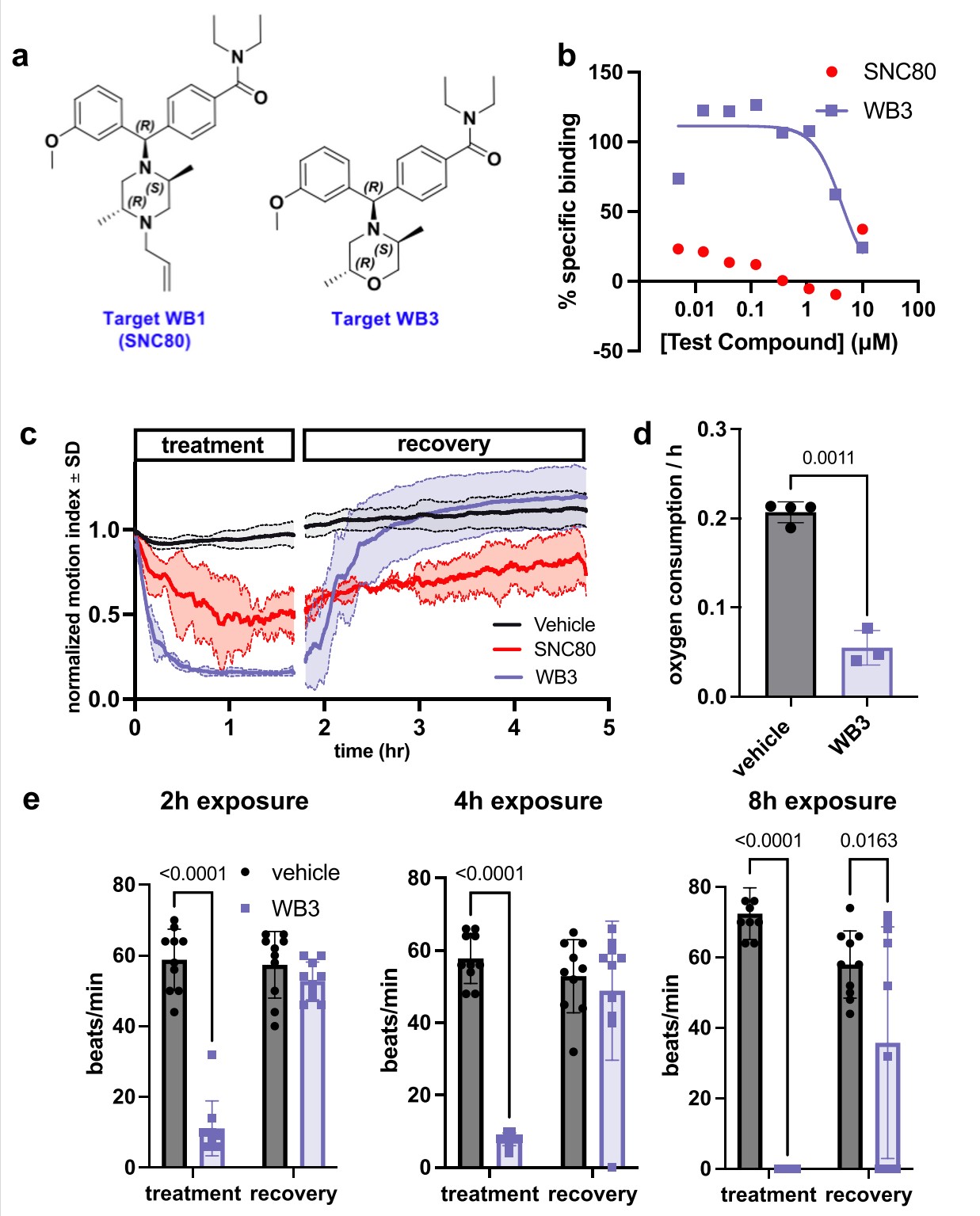

**Figure 3.** Design and in vivo screening of novel analog WB3. (**a**) Molecular structures for SNC80 and novel compound WB3. (**b**) Percent-specific binding of the delta opioid receptor radioligand [3H]-DADLE in the presence of SNC80 and WB3. (**c**) Swimming activity of *Xenopus* during 100 μM WB3, SNC80, or vehicle treatment. Data represent the mean ± SD of n=2 replicates, with 10 tadpoles per replicate. (**d**) Oxygen consumption rate for *Xenopus* treated with WB3 or vehicle controls. N=4 vehicle and n=3 WB3, with each data point representing the cumulative oxygen consumption for 5 tadpoles. (**e**) Heart rate in WB3 and vehicle-treated *Xenopus* (n=10 tadpoles/group). Statistical comparisons were performed using Welch's unpaired t-test (**d**) and a two-way ANOVA (group × timepoint) with Sidak's correction for multiple comparisons (**e**). Bar plots show the mean ± SD of each group.

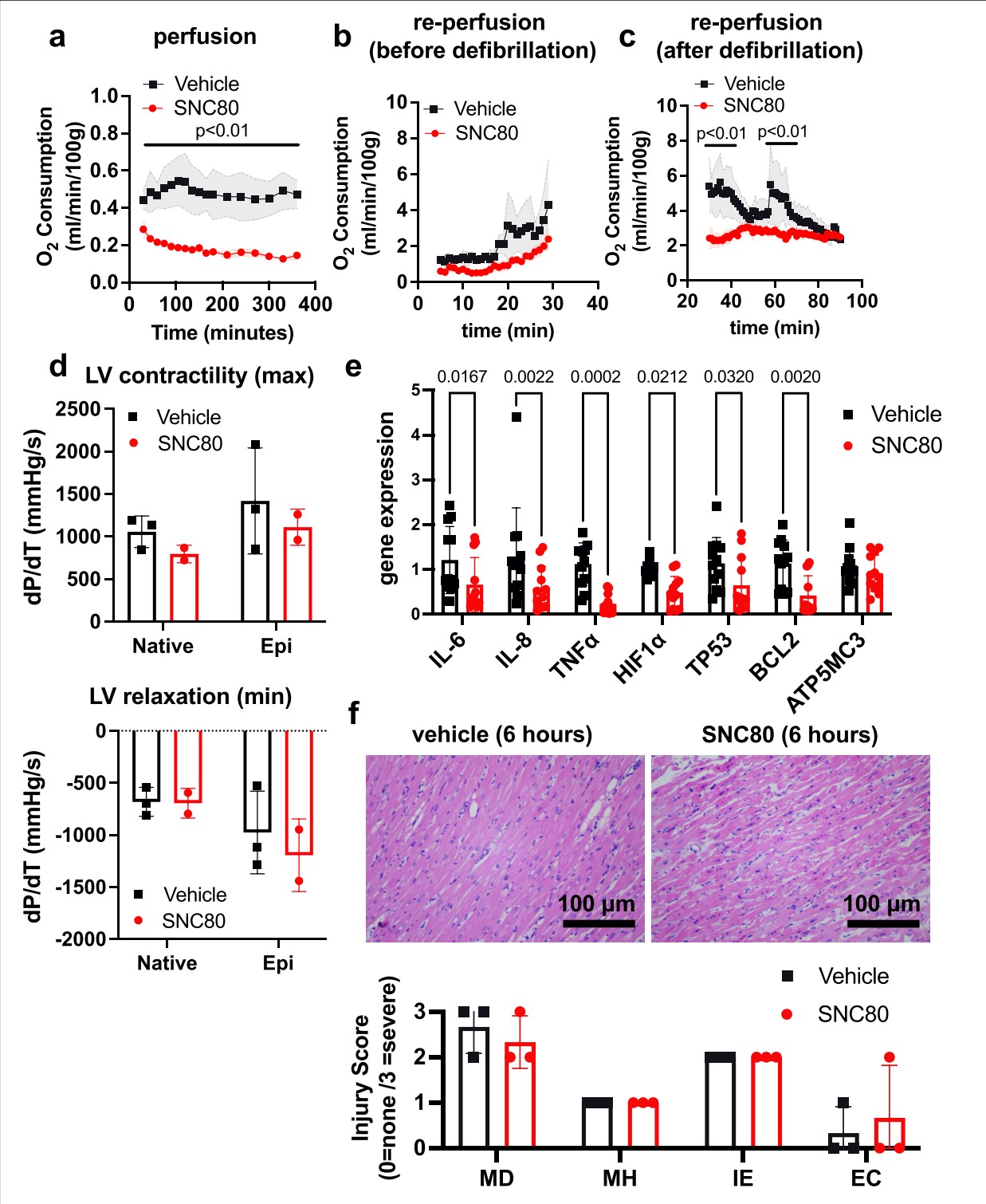

**Figure 4.** Preservation of porcine cardiac grafts with SNC80 over 6 hr. (**a**) Oxygen consumption for vehicle and SNC80-treated cardiac grafts perfused in the VP.S ENCORE device (n=3 hearts/group). (**b**) Oxygen consumption during cardiac reperfusion on the Langendorff system before defibrillation (n=3 vehicle and n=3 SNC80 hearts). (**c**) Oxygen consumption measured on the Langendorff system after defibrillation (n=3 vehicle and n=3 SNC80 hearts). (**d**) Left ventricle (LV) contractility and relaxation before and after defibrillation and tissue exposure to epinephrine (n=3 vehicle and n=2 SNC-treated hearts). (**e**) Gene expression analysis of heart biopsy samples for vehicle and SNC80-treated cardiac grafts (n=2 hearts/group; 3 technical replicates each). (**f**) Light microscopic views of hematoxylin and eosin (H&E)-stained histological sections of heart biopsies (×20 magnification) after 6 hr of perfusion and quantification of myocardial degeneration (MD), myocardial hemorrhage (MH), interstitial edema (IE), and endothelial changes (EC).

*Figure 4 continued on next page*

*Figure 4 continued*

Statistical comparisons were performed using multiple unpaired t-tests between SNC80 and vehicle-treated cardiac grafts (**a,b,c**), two-way ANOVA with Sidak's correction for multiple comparisons between SNC80 and vehicle-treated cardiac grafts (**d,f**), and unpaired t-tests between SNC80 and vehicle for each gene of interest (**e**). Data represent the mean ± SD.

The online version of this article includes the following figure supplement(s) for figure 4:

**Figure supplement 1.** Preservation of whole porcine limbs with SNC80.

of limb reimplantation due to complications from rhabdomyolysis, necrosis, and reperfusion injury (*Kaltenborn et al., 2020*). Again, we found that the relative metabolic rate measured during 3 hr of SNC80 exposure was significantly lower compared to vehicle-treated limbs (*Figure 4—figure supplement 1a*). Histopathological evaluation also showed retention of normal muscle morphology with no apparent changes in muscle bundle area (*Figure 4—figure supplement 1b*) or glycogen content (*Figure 4—figure supplement 1c*), suggesting that the tissue's potential for ATP synthesis was not compromised after SNC80 administration, although an increase in intercellular spacing was observed (*Figure 4—figure supplement 1d*). We did not observe full recovery of $O_2$ uptake at the 3 hr time-point analyzed in this study (*Figure 4—figure supplement 1c*); however, the porcine limb contains a large amount of fat tissue that may be acting as a depot for the highly lipophilic SNC80, thereby leading to extended stasis under these conditions.

## Elucidating potential SNC80 targets and pathways

To gain insight into the molecular basis of SNC80's hypometabolic and other physiological effects, we used thermal proteome profiling (TPP) to assess drug-protein binding and thereby identify previously

**Table 1.** SNC80 protein binding targets identified by thermal proteome profiling.

| Direct interaction | Function | Indirect interaction | Function |
|---|---|---|---|
| CCBL1 | Pyridoxal phosphate binding<br>Cysteine-*S*-conjugate beta-lyase activity<br>L-Glutamine aminotransferase activity<br>Cellular modified amino acid metabolic process | ABCB1 | ABC-type bile acid transporter activity<br>Regulation of fatty acid beta-oxidation<br>Efflux transmembrane transporter activity |
| EAAT1 | L-Glutamate transmembrane transporter activity | ACOX3 | Fatty acid beta-oxidation using acyl-CoA oxidase<br>Fatty acid binding |
| FUCA2 | Alpha-L-fucosidase activity<br>Glycoside catabolic process | CAMSAP2 | Cytoplasmic microtubule organization<br>Calmodulin binding |
| HM13 | Peptidase activity<br>Aspartic endopeptidase activity | COX6C | Mitochondrial electron transport, cytochrome *c* to oxygen<br>Generation of precursor metabolites and energy |
| PTX3 | Negative regulation of glycoprotein metabolic process<br>Complement component C1q complex binding | ITGA7 | Integrin-mediated signaling pathway<br>Cell adhesion mediated by integrin |
| RPE65 | Phosphatidylserine binding<br>Retinoid metabolic process | NCX1 | Calcium:sodium antiporter activity<br>Calcium ion binding<br>Regulation of the force of heart contraction |
| RPL27A | Structural constituent of ribosome<br>Cytoplasmic translation | ROBO1 | Roundabout signaling pathway<br>Aorta development |
| RPL4 | Structural constituent of ribosome<br>Cytoplasmic translation | RRP9 | RNA binding<br>rRNA processing |
| RPS25 | Structural constituent of ribosome<br>Cytoplasmic translation | | |
| TOMM40 | Protein transmembrane transporter activity<br>Protein targeting to mitochondrion<br>Mitochondrial outer membrane translocase complex | | |
| VMP1 | Autophagosome membrane docking<br>Plasma membrane phospholipid scrambling<br>Lipoprotein transport | | |

unknown molecular targets of SNC80 that might be critical for its metabolism slowing activity. TPP of SNC80-treated *Xenopus* tissues revealed 19 direct and indirect binding targets (*Table 1*), including proteins involved in transmembrane transport, mitochondrion activity, and metabolic processes. Two potentially interesting binding targets of SNC80 are the excitatory amino acid transporter 1 (EAAT1, also known as SLC1A3) and the $Na^+/Ca^{2+}$ exchanger 1 (NCX1, also known as SLC8A1), both of which have been identified in cardiac tissue (*Ralphe et al., 2004*; *Maiolino et al., 2017*). In addition to their individual functions, mitochondrial ATP production has been previously reported to be stimulated through intermolecular interactions between EAATs and NCXs in neurons (*Magi et al., 2013*). If this complex is also present in cardiac tissue, inhibition of EAAT and NCX1 could reduce cellular ATP levels (*Magi et al., 2019*), and thereby play a role in slowing metabolism. Specifically, the blockade of $Na^+/Ca^{2+}$ exchange via interference with EAAT1-NCX1 interactions could prevent the influx of calcium into the mitochondria that is necessary for calcium-sensitive mitochondrial dehydrogenases that fuel ATP synthesis (*Magi et al., 2019*). However, the mitochondrial variant of NCX1 (NCLX) was not included in the *Xenopus* protein library used for TPP analysis so could not be fully evaluated in this study. TPP also identified two additional SNC80-binding proteins located within the mitochondrion: TOMM40 and COX6C. TOMM40 is a translocase located in the outer membrane of the mitochondria that has been implicated in tissue protective effects against late-onset Alzheimer disease (*Chiba-Falek et al., 2018*) and COX6C is the terminal enzyme of the mitochondrial respiratory chain, catalyzing the electron transfer from reduced cytochrome *c* to oxygen.

In addition to interaction with mitochondrial proteins, we detected binding between SNC80 and enzymes involved in other metabolic processes, including alpha-L-fucosidase.

## Stasis induction in cultured human cells and tissues

To explore whether this drug also induces a state of biostasis in cultured human cells, we monitored the effects of SNC80 on oxygen consumption in four different types of cultured human cells – human umbilical vein endothelial cells (HUVECs), Caco-2 intestinal epithelial cells, liver sinusoidal endothelial cells (LSECs), and Huh7 liver epithelial cells – and analyzed multiple indicators of cellular metabolism using the Seahorse instrument. The latter allows to measure ECAR (extracellular acidification rate), typically serving as proxy for glycolysis activity and oxygen consumption rate (OCR) as a proxy for OXPHOS activity (since mitochondria are responsible for 70–90% of total oxygen consumption) (*Yang et al., 2020*). The Seahorse can also measure other mitochondrial respiration metrics such as ATP production.

Treatment with SNC80 (100 µM) suppressed OCR and ATP production relative to control levels within 1.5 hr in epithelial cells derived from intestine and liver (*Figure 5—figure supplement 1a*) without detectable cytotoxicity (*Figure 5—figure supplement 2a*), indicating a decrease in cellular metabolism similar to *Xenopus*. Furthermore, we observed a reduction in creatine kinase activity in both epithelial cell types (*Figure 5—figure supplement 2b*), indicating a decrease in overall cell energy demand (*Klepinina et al., 2021*).

However, cells can also compensate for decreased mitochondrial ATP production by activating other metabolic pathways (*Mookerjee and Brand, 2015*), indicated by an increase in ECAR. Typically, acidification is linked to lactic acid production due to glycolytic activity. Sometimes, however, it can be derived from cytosolic reductive carboxylation (CRC) liberating $NAD^+$ (nicotinamide adenine dinucleotide) and $H^+$. We used effluent glutamate levels as an indicator of the metabolic switch to glutamine utilization in CRC – a compensation typically used by cells with impaired mitochondrial respiration as a means for NADH recycling (*Gaude et al., 2018*).

The Caco-2 cells exposed to SNC80 showed a dramatic increase in ECAR and glutamate production (*Figure 5—figure supplement 2b*), suggesting that the cells activate compensatory mechanisms such as glycolysis and CRC pathway (*Fendt et al., 2013*; *Posho et al., 1998*). Glutamate generation can account for up to 40% of glutamine utilization in Caco-2 (*Posho et al., 1998*), and it is known that glutamine is a very important metabolic fuel and biosynthesis precursor for intestinal epithelium (*Costa et al., 2000*). We can also note that the cancerous origins of Caco-2 give them the ability to rapidly use glycolysis as a source of energy, commonly known as the Warburg effect (*Klepinina et al., 2021*). Unlike Caco-2, the Huh7 cells showed no change in ECAR nor glutamate (*Figure 5—figure supplement 2b*) after SNC80 treatment, suggesting that cells did not compensate mitochondrial impairment with increased glycolytic activity or CRC. SNC80 must therefore decrease total ATP

production, which was confirmed by detection of a significant drop in the ATP/ADP ratio within liver epithelial cells expressing a fluorescent reporter (*Figure 5—figure supplement 1b*).

When looking at endothelial cells, 2 hr of SNC80 treatment in HUVEC and LSEC reduced mitochondrial ATP production (*Figure 5—figure supplement 1a*), indicating a decrease in mitochondrial activity and slowing of metabolism. However, it did not alter OCR. We hypothesize that because endothelial cells are highly glycolytic (*De Bock et al., 2013*) with a low baseline metabolic activity (data not shown), that SNC80's effect on mitochondrial respiration does not change gut endothelial cell oxygen consumption. HUVEC and LSEC showed no change in ECAR or glutamate (*Figure 5—figure supplement 2b*), suggesting that the cells are not compensating with additional glycolysis or CRC and that therefore the overall ATP production in the cell line is decreased.

Taken together, these results indicate that all four human cell types showed significant decreases in mitochondrial metabolism in response to the biostasis inducer SNC80. However, Caco-2 cells were the only ones that activated compensatory pathways in response to the impaired mitochondrial respiration, while others just allowed for their energy balance to decrease significantly. Our studies also suggested that the metabolism-suppressing effects of SNC80 were most potent in the epithelial cells that were originally derived from highly metabolic tumors (*Wan et al., 2022*).

## Metabolic suppression in human Organ Chips

To explore whether this drug also induces a state of biostasis in human cells and tissues, as well as non-cardiac cells, we tested SNC80 in human organ-on-a-chip (Organ Chip) microfluidic culture devices that reconstitute tissue-tissue interfaces and human organ-level physiology with high fidelity in vitro (*Ingber, 2022*). The Organ Chip models reconstitute a tissue-tissue interface between organ-specific epithelium and endothelium across a porous extracellular matrix (ECM)-coated membrane within a two-channel microfluidic device, and hence allow tissue-tissue cross-talk observed in vivo that is not

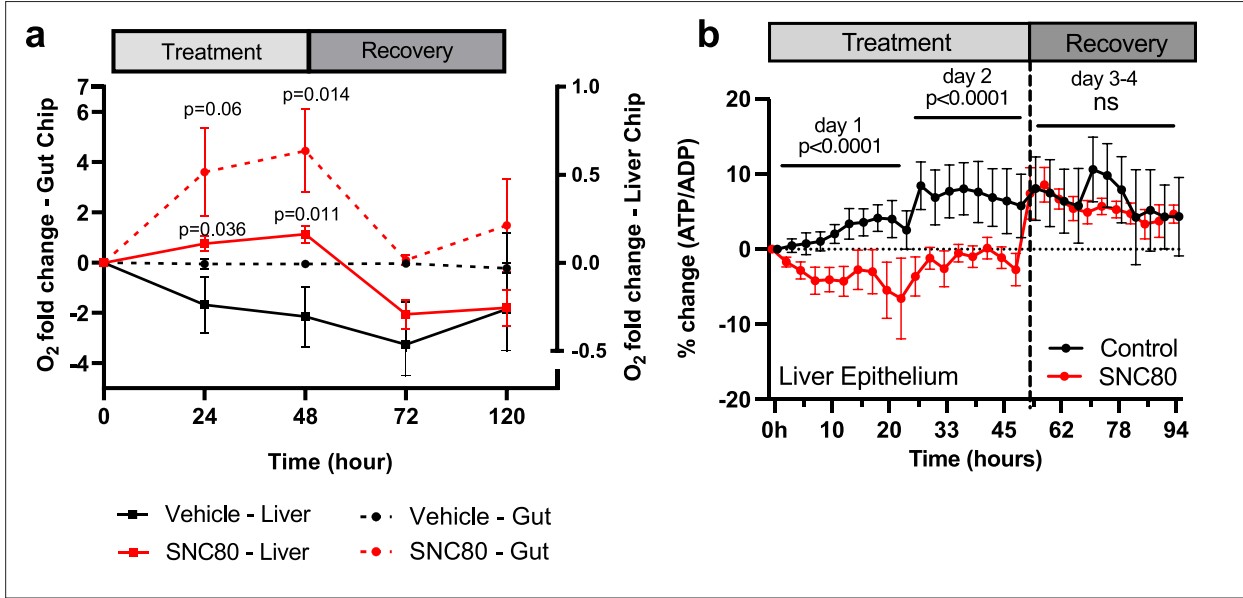

**Figure 5.** SNC80 treatment in human Organ Chips. (**a**) Monitoring changes in $O_2$ levels in human Gut (dotted lines; n=6 Chips/group) or Liver (straight lines; n=15–16 Chips/group) Chips throughout the treatment with 100 µM SNC80 (0–48 hr) and recovery (48–120 hr) versus vehicle control chip. (**b**) Graph showing the quantitative drop in ATP/ADP ratio of Liver Chip treated for 48 hr with 100 µM SNC80 or vehicle control as well as 2 days recovery after drug washout (n=5 control and n=3 SNC80 Chips). Statistical comparisons were performed using multiple unpaired t-tests for each timestamp with multiple comparison between control and treated cells. Data represent the mean ± SD.

The online version of this article includes the following figure supplement(s) for figure 5:

**Figure supplement 1.** SNC80 treatment in cell culture and Organ Chip systems.

**Figure supplement 2.** Metabolic measurements in gut and liver cultures treated with SNC80.

**Figure supplement 3.** Characterization of human Gut and Liver Chips with integrated sensing technology.

**Figure supplement 4.** Metabolic measurements of Gut and Liver Chip systems treated with SNC80.

normally present in studies with cultured cells, as previously described (*Si et al., 2021*; *Kim et al., 2016*). In this study, we used a Gut Chip lined by Caco-2 intestinal epithelial cells interfaced with HUVECs and a Liver Chip containing Huh7 liver epithelial cells interfaced with LSECs, respectively. However, we modified the Organ Chip design by integrating sensors that allow for live monitoring of oxygen ($O_2$) and transepithelial electrical resistance (TEER) (tissue barrier function) (*Figure 5—figure supplement 3a*). Both Organ Chips were treated with SNC80 by flowing the drug in medium for 2 days through the apical and basal channels of the chip, and then removing the drug followed by 7 days of additional culture. We confirmed that the intestinal epithelial cells underwent villus differentiation and accumulated mucus under these culture conditions (*Figure 5—figure supplement 3b*), in addition to maintaining a low oxygen environment after 10–16 days of culture (*Figure 5—figure supplement 3c*) as previously described (*Si et al., 2021*; *Kim et al., 2016*). The Liver Chips also produced albumin and urea at levels (5 and 135 µg/day/chip, respectively) (*Figure 5—figure supplement 3d and e*) that were similar to those measured in Organ Chips lined by primary hepatocytes (*Choi et al., 2016*; *Ewart et al., 2021*).

These studies revealed that SNC80 treatment reduced the respiratory (oxygen utilization) rate in the Gut Chip, resulting in an almost sixfold increase in extracellular oxygen levels after 48 hr, which then progressively returned to baseline levels during the 7-day washout period (*Figure 5a*). Similar results were obtained with the more highly metabolic Liver Chip with SNC80 inducing higher levels of extracellular oxygen compared to control chips and this again reversed after drug washout (*Figure 5a*). Importantly, the tissues in both the human Organ Chips remained healthy after 2 days' treatment with SNC80 as there was no significant change in barrier function (*Figure 5—figure supplement 4a and b*), cell growth (*Figure 5—figure supplement 3g*), or cytotoxicity (*Figure 5—figure supplement 4c and d*). Similar to the data observed in cell cultures, ATP/ADP ratio in Liver Chips was significantly decreased after 24 and 48 hr of SNC80 treatment compared to the vehicle control chips. However, ATP/ADP ratio measurements between SNC80 and vehicle control were not significantly different after washout of the drug, indicating that Liver Chip tissues recover their normal metabolic function up to 4 days after treatment (*Figure 5b*).

Collectively, these data suggest that treatment with SNC80 in these physiologically relevant human cells and tissues cultured on-chip in an organ-relevant context resulted in a drop in respiration, with no switch toward glycolysis or reductive carboxylation, since the levels of lactate (*Figure 5—figure supplement 4e and f*) and glutamate remained unchanged in both chips (*Figure 5—figure supplement 4g and h*). Thus, in this organ-like microenvironment, the tissues decrease their total ATP production in presence of SNC80, which is accompanied by a global slowing of metabolism.

## Discussion

Here, we report the discovery that the small molecule drug, SNC80, and its analog WB3, are capable of slowing of multiple physiological processes, and thereby inducing a hypometabolic state without external cooling. Although there is extensive clinical experience using hypothermia to cryopreserve organs during surgery (e.g. cardiac bypass) and for donor transplants, comparatively little work has been done in the area of molecular induction of biostasis without external cooling. We found that SNC80 reversibly suppressed metabolism across multiple cell and tissue types, as well as different species and experimental models, at their basal temperatures, suggesting that this approach could be appropriate for preservation of cells and tissues of the heart, liver, and gut. Critically, integrating this small molecule drug in a commercial organ preservation system with whole explanted pig hearts demonstrated potential for near-term translation and impact of this approach for applications relating to organ preservation and surgical transplantation.

Initially developed as a nonaddictive pain reliever, SNC80 was previously shown to induce hypothermia and protect against the effects of spinal cord ischemia in rodents (*Rawls et al., 2005*; *Horiuchi et al., 2004*), but its broad effects on metabolism were not known. In the present study, treatment with SNC80 slowed swimming movement and reduced OCR in a reversible manner in *Xenopus* tadpoles, and it lowered the metabolic rate in non-mobile *Xenopus* embryos as well. Increased levels of acylcarnitine and cholesterol ester in the SNC80-treated *Xenopus* suggested a specific reduction in fatty acid metabolism activity, similar to the metabolic changes observed in hibernating brown bears and ground squirrels (*Soeters et al., 2009*; *Welinder et al., 2016*; *Otis et al., 2011*). Importantly, SNC80 displayed similar metabolism suppressing activities that were reversible in physiologically relevant

human Organ Chip microfluidic devices that reconstitute organ-level structures and functions in vitro. These experiments also aided in the dose selection for SNC80 (100 μM), which was chosen to maximize suppression of metabolic and activity parameters, while ensuring reversibility of biostasis. These data suggested that induction of a hypometabolic state by this small molecular drug could potentially provide a new paradigm to prolong organ preservation. Indeed, when we perfused explanted porcine hearts and limbs with SNC80, we experimentally confirmed that this biostasis agent can slow metabolism in a whole complex living organs, providing direct support for its potential integration into commercial organ preservation devices that strive to extend the viability of human organs for surgical transplantation.

Similar to molecules assessed in the present study, hydrogen sulfide has been shown to induce hypometabolism in whole organs at their basal temperature (*Maassen et al., 2019*). Hydrogen sulfide perfusion in porcine kidneys reduced oxygen consumption with similar strength as SNC80 across assays, but its activity was less pronounced than WB3. Notably, hydrogen sulfide did not impact ATP levels in porcine kidneys, whereas SNC80 suppressed ATP production in multiple 2D cell assays and liver epithelium of the Liver Chip. SNC80 is defined as a highly selective non-peptide DOR agonist drug and other DOR agonists, such as the enkephalin DADLE, have been shown to reduce metabolic demand and protect against oxidative stress, thereby decreasing injury and apoptosis in perfused organs (*Dou et al., 2022*; *Beal et al., 2019*; *Ratigan and McKay, 2016*). Similar to the effect of SNC80 in porcine hearts, DADLE decreased markers of tissue injury and apoptosis in the rat liver (*Beal et al., 2019*), However, DADLE increased ATP levels in liver and increased oxygen consumption in hepatocytes exposed to oxidative stress, suggesting distinct mechanisms compared to SNC80.

Prior studies of DADLE found that the DOR pathway was required to alter metabolism and provide protective effects, however, we found that SNC80 suppresses metabolism via a mechanism that is independent of its DOR modulating activity. Instead, we found that SNC80 interacts with multiple proteins involved in transmembrane transport, mitochondrion activity, and metabolic processes. Interestingly, one of the identified targets, NCX1, has been previously implicated in control of temperature compensation and autonomous oscillation (*Kon et al., 2021*), suggesting that it plays a key role in the regulation of biochemical reaction speed, cell cycle, and other forms of biological timing, such as circadian rhythm. In addition to its effects on tissues of the nervous system, multiple drugs that target $Na^+/Ca^{2+}$ exchange by NCX, such as the KBR family of drugs, were developed for the treatment of ischemia, reperfusion injury, and heart arrhythmias (*Shah Amran et al., 2003*). In rats, pre-ischemic or post-ischemic treatments with KBR compounds improved recovery of ventricular pump function and reduced reoxygenation-induced injury as measured the reduced release of intracellular components (e.g. lactate dehydrogenase) (*Shah Amran et al., 2003*; *Schäfer et al., 2001*). Our results suggest that SNC80 could be acting on the NCX1 transmembrane protein and/or NCX1/EAAT1 pathway may result at least in part from induction of a hypometabolic state (*Ingber, 2022*). Of the tissues impacted by SNC80, the observed effects were most profound on *Xenopus* tadpole heart function and porcine hearts. SNC80 and other opioids have been shown to decrease cardiac inotropy and contractility (*Headrick et al., 2015*; *Pugsley, 2002*), and prior work has demonstrated that decreases in cardiac inotropy and contractility by opioid agonists may be mediated by opioid receptor-*independent* actions of these drugs (*Pugsley, 2002*). Substantial slowing of heart rate by WB3, which exhibits approximately 1000 times less delta opioid activity than SNC80, further supports this claim. Similar to our findings by TPP, other investigators identified *non-opioid* receptor-mediated actions of these compounds on ion channel function (*Pugsley, 2002*). Changes in contractility were not observed immediately after perfusion with SNC80 in porcine hearts tested here, likely because the SNC80 was washed out of the tissue before contractility was measured; however, future work should continue to monitor these parameters, especially at later timepoints. Although bradycardia likely contributes to the hypometabolism observed in both systems, we also see reduced metabolic rates in systems lacking cardiac cells, and thus the hypometabolic activity of SNC80 we observed is not completely cardiac-specific.

The *Xenopus* and perfused porcine heart systems may be particularly amenable to extreme slowing of cardiac output because these live tissues are surrounded by well-oxygenated medium during treatment, potentially avoiding tissue injury sometimes associated with hypometabolic states. However, the assumption that high oxygen content in the perfusate completely precludes ischemia may not be valid. Assuming adequate flushing during recovery, local vasospasm or a fat microembolism occurring

during perfusion could restrict perfusion flow thereby inducing a localized hypoxic state and HIF1α expression in the tissue. In addition, although HIF1α is primarily known for its role in response to hypoxia, it has some oxygen-independent functions and can be induced by factors other than oxygen levels, such as growth factors and cytokines, which can activate its transcriptional activity even under normoxic conditions. In these cases, HIF1α can regulate genes involved in cell proliferation, metabolism, inflammation, and other cellular processes. Therefore, observation of suppressed HIF1α, as well as the reduction of multiple inflammatory and cell death markers, in the treated porcine cardiac tissue suggests a protective effect of SNC80.

Before considering broader applications of SNC80 or any of its analogs, such as whole-body infusion during trauma or infection, it will be critical to consider impacts of this drug not only on the heart but other organ systems as well. In particular, our *Xenopus* data suggest that there could be substantial uptake of SNC80 in the brain, thus potentially impacting tissues of the central nervous system. Therefore, future work should characterize cognitive recovery in organisms receiving systemic delivery of stasis inducers. In addition, whole-body infusion applications may require further pharmaceutical development, including chemical modifications to SNC80 and development of targeted delivery methods to ensure clinical safety.

In summary, our findings demonstrate that the small molecule drugs SNC80 and WB3 can be used as a biostasis inducer to produce a hypometabolic state, which can provide potential therapeutic value for clinical applications, such as enhanced preservation of organs. The use of a biostatic compound in combination with a self-contained transport system could potentially increase organ viability for sustained times ex vivo, thus helping to advance organ and tissue preservation for surgical transplantation.

## Methods
### *Xenopus* model

*X. laevis* (wild-type) embryos were fertilized at Tufts University using procedures reviewed and approved by the Tufts University Institutional Animal Care and Use Committee regulations (M2020-35) and transferred to the Wyss Institute. *Xenopus* embryos and tadpoles were housed at 18°C with a 12/12 hr light/dark cycle in 0.1X Marc's Modified Ringer's medium. All animal experiments and procedures were reviewed and approved by the Harvard Medical School (HMS) Institutional Animal Care and Use Committee regulations (IS00000658-6). *Xenopus* that exhibited signs of poor health prior to treatments were excluded from studies, including abnormal development or poor mobility. *Xenopus* experiments did not identify male versus female samples and therefore a random sampling was used.

Drugs used in the *Xenopus* studies were dissolved in DMSO to a stock concentration of 10 mM and diluted to their final concentrations with a 1% DMSO concentration or less. Media with dosed compound were made fresh and screening was performed in tadpoles at stages 46–50. The activity of free-swimming tadpoles during and after exposure to drugs were recorded in 60 mm dishes using a SONY Alpha a6100 camera with 16 mm objective (Sony Corporation, Tokyo, Japan) against an illuminated background. Mobility was quantified in MATLAB (Mathworks, Natick, MA, USA) by mapping differences between frames to a movement index (0–1), with 0 indicating no movement and 1 indicating maximal movement. Oxygen consumption was measured during *Xenopus* exposure to drugs as a proxy for whole-organism metabolism using the Firesting Optical Oxygen Meter and the Pyro Oxygen Logger software (v3.317) (Pyroscience GmbH, Aachen, Germany). If leaks or bubbles were detected in Firesting oxygen vials during an experiment, acquired data were not used in the analysis. Oxygen consumption for *Xenopus* embryos was measured using the OxoPlate (PreSens Precision Sensing GmbH, Regensburg, Germany). OCR for each sample was measured using a linear fit to the oxygen consumption curves in Prism (GraphPad, San Diego, CA, USA). Heart rate was measured by microscopy imaging in *Xenopus* following 1, 2, 4, or 8 hr of drug exposure. Vehicle and drug-treated groups were briefly incubated in 0.01% tricaine (Syndel, Ferndale, WA, USA) to minimize tadpole movement during heart imaging. Videos of heart function were acquired using a ZEISS Axio Zoom. V16 microscope and ZEN BLUE Microscopy software (v3.1) (Carl Zeiss AG, Oberkochen, Germany) at a rate of 1 frame/100 ms. Heart beats were counted automatically using a custom code in Python. At the completion of experiments, *Xenopus* were euthanized by immersion in 0.2% tricaine for 30 min, followed by fixation or bleaching and disposal.

## MALDI-ToF

Tadpoles were embedded in gelatin (0.11 g/mL) and stored in –80°C. Embedded tadpoles were sectioned into 16-μm-thick slices using a Leica CM1850 (Leica Microsystems, Wetzlar, Germany) onto indium tin oxide (ITO)-coated glass slides (Bruker Daltonics, Billerica, MA, USA) and immediately placed in a vacuum desiccator before matrix application (N=5 tadpoles per condition; N=3 sections/ slide). Experiments were duplicated to confirm the reproducibility. The matrix solution was composed of 2,4-dihydroxybenzoic acid (40 mg/mL) in 0.1% formic acid, methanol/water (1:1 vol/vol) and was made fresh for each application. The HTX-sprayer nebulizer (HTX Technologies, Carrboro, NC, USA) was used to apply the matrix at 80 °C at 24 passes over the sample, at a flow rate of 50 μL/min, 10 psi pressure, and a track speed at 1250 mm/min. Before imaging, SNC80 spotting with matrix and *Xenopus* lysate was performed to determine the optimal matrix for the best S/N ratio and the least ion suppression from *Xenopus* tissue and to identify which adducts were detectable for each compound for imaging. In situ MALDI-ToF imaging was acquired using the rapifleX (Bruker Daltonics, Billerica, MA, USA), in positive ion mode at 1000 Hz in the mass range of 300–990 m/z. The laser raster step spacing was set to 45 μm step size at 500 shots per pixel. External calibration was performed using a red phosphorus slurry drop casted on a region of the ITO slide without tissue. Preliminary processing of the data was performed using FlexAnalysis (Bruker Daltonics, Billerica, MA, USA). Images were normalized by the total ion count and baseline-corrected using Top Hat algorithm with a _baseline width. After MALDI-ToF MSI acquisition, ITO slides were then stained for hematoxylin and eosin (H&E) and imaged using the Cytation 5 (Agilent, Santa Clara, CA, USA) and Epsilon scanner at 3000 dpi to select regions of interest using the open-source program, Qupath (*Bankhead et al., 2017*). Mass selection windows and signal intensity analysis were conducted with SCiLS Lab (Bruker Daltonics, Billerica, MA, USA), where ions of interest, including SNC80, acylcarnitine, and cholesterol ester C18, were chosen with a width of ±0.25 Da. Reference peaks of target analytes were based on previous MALDI-ToF work in *Xenopus* tissues from previous findings (*Goto-Inoue et al., 2016*), and during spotting of target analytes with MALDI-ToF MS/MS. The mean signal intensity of the defined regions of interest was obtained with each ion of interest.

## Radioligand binding assay

To assess binding of WB3 to the DOR, we tested WB3 and SNC80 at 8-concentrations with three-fold serial dilutions starting at 10 μM (Reaction Biology, Malvern, PA, USA). Binding of the tracer [3H]-DADLE (10 nM) was assessed in the presence of each concentration of WB3 and SNC80 and the [3H]-DADLE signal was fit to a dose-response curve to quantify the Hill slope and IC50 values.

## Thermal proteome profiling

To identify drug-bound targets based on altered thermal stability, *Xenopus* tadpoles were treated with either SNC80 or vehicle for 2 hr. *Xenopus* were euthanized in 20× tricaine and then transferred back to their respective drug solutions to ensure drug effect is not lost during heat treatment. *Xenopus* samples were incubated at nine temperature points (30°C, 35°C, 40°C, 44°C, 48°C, 52°C, 56°C, 60°C, 65°C) using a PCR machine (3 min heat time per condition, 3 tadpoles per condition) for both drug-treated and vehicle-treated groups. After transfer to room temperature, liquid was aspirated from samples and samples were snap-frozen in liquid nitrogen. Samples were lysed in 100 μL of PBS+0.4 % NP-40 with 4 cycles of freeze/thaw. 11 μL of each of the nine temperatures were pooled in one tube and samples were centrifuged at 13,000 RPM for 75 min, at 4°C. Proteins were reduced and alkylated with 10 mM DTT and 15 mM IAA for 15 min at 65°C and 30 min at room temperature, respectively. Proteins were then precipitated with 8 volumes of ice-cold acetone and 1 volume of ice-cold methanol overnight. Digestion was carried for 4 hr in 50 mM Tris pH 8+0.75 mM urea+1 μg trypsin/LysC at 37°C with agitation. Another 1 μg of trypsin/LysC was added and digestion was continued overnight. Peptides were purified by reversed phase SPE and analyzed by LC-MS/MS (PhenoSwitch Bioscience, Sherbrooke, QC, Canada). LC-MS/MS acquisition was performed with a ABSciex TripleTOF 6600 (ABSciex, Foster City, CA, USA) equipped with an electrospray interface with a 25 μm i.d. capillary and coupled to an Eksigent μUHPLC (Eksigent, Redwood City, CA, USA). Analyst TF 1.8 software was used to control the instrument and for data processing and acquisition. Samples were acquired twice in SWATH acquisition mode using gas phase fractionation. The source voltage was set to 5.5 kV and maintained at 325°C, curtain gas was set at 45 psi, gas one at 25 psi and gas two at 25 psi. Separation

was performed on a reversed phase Kinetex XB column 0.3 µm i.d., 2.6 µm particles, 150 mm long (Phenomenex) which was maintained at 60°C. Samples were injected by loop overfilling into a 5 µL loop. For the 60 min LC gradient, the mobile phase consisted of the following solvent A (0.2% vol/vol formic acid and 3% DMSO vol/vol in water) and solvent B (0.2% vol/vol formic acid and 3% DMSO in EtOH) at a flow rate of 3 µL/min. GPF files for both acquisitions were analyzed on a previously generated 3D ion library. Each GPF file was analyzed using 10 peptides per protein, 4 MS/MS transition per peptide, 12.5 min RT window, and 25 ppm XIC width in Peakview (Sciex). The reported quantification for a protein is the sum of all the correctly integrated peptides (FDR<0.05) in both GPF files.

## Human cell cultures

Caco-2 intestinal epithelial cells (Caco-2 BBE human colorectal carcinoma cells, Harvard Digestive Disease Center) were grown in high glucose (4.5 g/L) DMEM (Gibco), 10% FBS (Gibco), and 1% penicillin/streptomycin (Gibco) and subcultured every 2–3 days at 1:6 ratio. For all experiments, cells were used between passages 59 and 64. HUVECs (Lonza, cat#: C2517A) were maintained in EBM-2 (Lonza) media with EGM-2 BulletKit (Lonza), 2% FBS, and 1% penicillin/streptomycin (Gibco) and subcultured every 2–3 days at 1:6 ratio. For all experiments, cells were used between passages 3 and 7. HuH-7 liver epithelial cells (Perceval, cat#: HuH-7) were maintained in DMEM with 4.5 g/L glucose (HyClone), 10% FBS, and 1% penicillin/streptomycin (Gibco). We also obtained wild-type Huh7 transduced with lentivirus particle pLV[Exp]-Puro-CMV>{Perceval(ns)}:T2A:mCherry/3xNLS (Vector Builder) that were used for live imaging of changes in ATP/ADP ratio within these cells, which directly correlates to cellular metabolic activity (*Berg et al., 2009*). LSECs (Lonza, cat#: HLECP1) were maintained in EBM-2 (Lonza) media with EGM-2 BulletKit (Lonza), 2% FBS, and 1% penicillin/streptomycin (Gibco).

## Metabolic assays

Effluents were collected at different timepoints (baseline, 24 hr, 48 hr, and 1–3 days recovery after washout) throughout the exposure to SNC80 or 1:1000 DMSO vehicle. After treatment, the effluent was collected and analyzed for lactate and glutamate production using the Lactate-Glo (Promega, #J5021) and Glutamate-Glo (Promega, #J7021) assay kits respectively as per the manufacturer's protocols.

## Creatine kinase assay

Creatine Kinase Colorimetric Activity Assay Kit (abcam, #ab155901) was used as per the manufacturer's protocols. Creatine kinase activity in experimental cells was deduced from the colorimetric values obtained from the standard curve.

## Seahorse assay

Four assays were run successively, one for each cell type – Caco-2, Huh7, HUVEC, LSEC. Each cell type was plated on Agilent XFe24 plate to perform a Mitostress Assay. Cells were added in all wells except A3, B1, C4, and D2 used for background measurements. Cell concentrations were chosen to be 90% confluent on the day of experiment.

The days prior to experiment, the cartridge was hydrated in calibrant and incubated in non-CO$_2$ incubator overnight. The day of experiment, cell media was changed to Seahorse XF medium to which 1 mM pyruvate, 2 mM glutamine, and 10 mM glucose were added. Cells were allowed to stabilize in this medium at 37°C in a non-CO$_2$ incubator for 1 hr. In the meantime, the drug concentrations were prepared in Seahorse XF medium at a 10× concentration stock, so that once injected on the cells, the concentration is diluted to 1×. Stock values were 1 mM SNC80. 56 µL of the drug (or media with DMSO vehicle for control wells) were loaded in port A of the respective cartridge well. Port B was loaded with 62 µL of 10 µM oligomycin, port C with 69 µL of 10 µM FCCP, and port D with 76 µL of 5 µM antimycin A and 5 µM rotenone. The Seahorse instrument was programmed so that baseline OCR and ECAR measurements were taken before the drugs were injected. Drugs were incubated for 2 hr, another measurement was taken, and mitochondrial inhibitors were successively injected in the following order: oligomycin, FCCP, rotenone. With the acquired measurements, we used the Agilent Wave software to process the data.

## Cytotoxicity assay

Cytotoxicity measurements on the Organ Chip effluents were performed using the LDH-Glo (Promega, #J2380) as per the manufacturer's protocols.

## ATP/ADP ratio measurements

Huh7 cells were transfected with a reporter that enables direct measurements of cellular ATP:ADP ratio. The basal excitation spectrum of the reporter protein was similar to that of YFP (peak around 490 nm) but it had an additional peak at CFP wavelength (405 nm). As ATP levels decreased, more ADP can bind to the site, decreasing the YFP/CFP ratio value. BioTek Cytation 5 Cell Imager (Agilent) was used to take RFP, YFP, and CFP images of each well every 20 min for 180 min upon exposure to 100 µM SNC80 or 1:100 DMSO vehicle. Post-processing of the images was performed using the open-source CellProfiler program (*Stirling et al., 2021*). We implemented the following pipeline to obtain quantitative ATP/ADP ratio values: (1) Identify nuclei with RFP marker, (2) define a cell as 4pixels around the edge of the nuclei, (3) measure cells' YFP and CFP intensity and YFP/CFP ratio, (4) count cells per image, (5) track cells' YFP and CFP intensity over the time course (0–3 hr with 20 min intervals), (6) measure YFP/CFP ratio average **over all cells** per image and normalize to cell count (7).

For the qualitative images, raw images were processed with ImageJ by using the image calculator to divide YFP/CFP intensity of respective images. 'Rainbow RGB' lookup table was used to show the intensity changes. Tones toward the red show more intensity, higher ATP/ADP ratio, and higher metabolism.

## Organ Chips with integrated sensors

To allow for live oxygen and transepithelial/transendothelial electrical resistance (TEER) monitoring, we used our in-house developed and fabricated microfluidic chips with integrated TEER and oxygen sensors. To mimic the in vivo-like cell-cell interface on the models, a dual-channel PDMS microfluidic chip was used where the apical channel is separated from the basal channel with a porous PDMS membrane. A pair of TEER sensing electrodes were integrated on the apical side of the top channel and the basal side of the bottom channel to measure the electrical resistance across the epithelium-endothelium tissue interface. The $O_2$ sensing nanoparticles (OXNANO, PyroScience) were integrated at four different locations in the chip: two in the apical channel and two in similar locations in the bottom channel. This is to allow a fair representation of the whole chip oxygen tension levels.

After fabrication, the chips were sterilized by plasma treatment and desiccated for 30 min. The polymeric surfaces of the membrane were then chemically functionalized using ER1/ER2 buffers following Emulate Inc protocol and coated with ECM for 1.5–2 hr at 37°C. Gut Chip apical and basal channels were coated with 30 µg/mL Collagen I (Advanced BioMatrix, cat#: 5005) and 100 µg/mL Matrigel (BD Biosciences) in cold PBS. Liver Chip apical channel was coated with 30 µg/mL fibronectin (Corning, cat#: 356008) and 400 µg/mL Collagen I (Advanced BioMatrix, cat#: 5005) in cold DPBS, while the basal channel was coated with 100 µg/mL Matrigel (BD Biosciences) and 200 µg/mL type I collagen in cold DPBS. To create Gut Chip, HUVECs (6–8×10⁶ cells/mL (P5)) were first seeded on the bottom side of the permeable membrane in the basal channel and incubated for 2–3 hr at 37°C before seeding the Caco-2 cells (2–3×10⁶ cells/mL (P78)) on the membrane in the apical channel. Liver Chip was similarly created by seeding the LSEC (4–5×10⁶ cells/mL (P3–8)) the first in the basal channel followed by seeding the Huh7 hepatocytes (2.5–3.5×10⁶ cells/mL (P5–15)) on the membrane in the apical channel. The chips were then cultured under continuous flow using the commercial culture module instrument (ZOË Culture Module, Emulate Inc, USA). The Gut Chips were maintained under continuous flow of 30 µL/hr in the apical and basal channels for 10 days before changing their respective media to differentiation-promoting media (5% FBS in DMED apically and 0.5% FBS endothelial medium basally) for the rest of the culture time. The Liver Chips were also maintained under continuous flow of growth media at 30 µL/hr in the apical and basal channels for 5–6 days before reducing the FBS content to 5% in Huh7 media and 0.5% in LSEC media for inducing tissue differentiation.

Sensor chips that had fabrication imperfections (leaks, inconsistent background TEER measurements, bonding failure) were discarded and not used for cell loading. If leaks or contamination appeared on the chips after loading, before or during an experiment, the chip was discarded, and its acquired data were not used in the analysis. If RFP (nuclear reporter) signal was absent in the Liver Chip, suggesting poor cell viability, the chip was discarded.

## Human Organ Chip studies

SNC80 was tested on the Gut Chip at day 16 of culture, when the epithelium showed characteristics of matured, functional tissue and the Liver Chip was tested after 7–10 days of culture. SNC80 (100 µM)

in the culture media was perfused through the apical and basal channels of the chips at 30 µL/hr, while the control chips were perfused with the same culture media containing the compound's vehicle (0.5% DMSO) using the same flow condition. The chips were treated with SNC80 for 48 hr before the drug was removed from the media and the culture was continued for another 4 days. Both Liver and Gut Chips were monitored at 0 hr, 3 hr, 24 hr, 48 hr timepoints as well as days 1 and 3 after the drug washout for oxygen consumption, TEER measurements, and effluent analysis. Fluorescence microscopy was continuously performed from 0 hr throughout the treatment period and up to 3 days of recovery. CellProfiler program was used to analyze ATP/ADP ratio as described above.

## Oxygen and TEER measurements on-chip

Oxygen tension level measurements was performed on-chip during the culture using FireSting-O2 Optical Oxygen and Temperature Meter (FSO2-C4, Pyroscience Sensor Technology, Germany) connected to Pryo Oxygen Logger (V3.317, Germany). TEER measurements were performed on-chip using our in-house developed multiplexed TEER Tray and remotely measured from outside the incubator. A four-point impedance measurement was taken on-chip periodically every 24 hr during the SNC80 treatment and recovery using the IviumSoft application (V4.97, Informer Technologies, Inc). To obtain the TEER of only the epithelial-endothelial tissue interface, the measured impedance at 100 Hz was subtracted from the impedance at 100 kHz and reported as the tissue impedance ($\Omega$) throughout the culture and in response to different stimuli as we reported previously (*Henry et al., 2017*).

## DNA damage and proliferation assays

The Click-IT EdU cell proliferation kit (Thermo Fisher, #C10337) was used to assess cell proliferation in the formaldehyde-fixed chips and DNA damage was assessed by staining anti-H2AX antibody (Novus Bio NB100-2280).

## Urea assay

Urea nitrogen was measured using a colorimetric assay (urea nitrogen test, Stanbio Laboratory), whereby undiluted Liver Chip effluents were added to a 96-well plate and mixed with the kit's working reagents as per the manufacturer's protocol.

## Albumin assay

Liver albumin production was measured using an enzyme-linked immunosorbent assay (ELISA) (Bethyl Laboratories, E88129). Liver Chip effluents were diluted 1:100 in buffer before being plated in the kit's 96-well plate, aside from standard wells with dilution series of known albumin concentration solution. We followed the manufacturer's protocol to perform the sandwich ELISA and measure absorbance at 450 nm.

## Porcine heart ex vivo perfusion studies

After induction of anesthesia and preparation for clean, non-survival surgery, a median sternotomy was performed followed by longitudinal incision in the pericardium of male and female Yorkshire pigs to expose the heart and the great vessels. Following blunt dissection and exposure of the vena cava, aorta and the brachycephalic artery, the superior vena cava, and azygos vein was ligated and divided. The brachycephalic artery was catheterized for infusion of cardioplegia solution. The inferior vena cava was clamped and divided proximal to the clamp. Two or three pulmonary arteries were incised for decompression of the left side of the heart. The aorta was immediately clamped and 10–15 mL/ kg of body weight of cold (2–4°C) cardioplegia solution (del Nido without lidocaine, 3 L) was infused at a pressure of 70–80 mmHg to stop the heart. Cold 4°C saline and ice were poured over the heart and the accumulated fluids from the thoracic cavity were removed by suction. Following cessation of myocardial function, the heart was removed from the chest by further dissection and division of the pulmonary vessels and aorta distal to the clamp. The heart was cooled in a surgical bowl with cold saline for inspection and preparation for perfusion. All procedures were conducted under the UT Health San Antonio approved Institutional Animal Care and Use Committee (IACUC) protocol 20210080AR.

After the heart was removed and placed in a surgical bowl, the aorta and the coronary sinus were cannulated using sterile zip ties. The heart was placed into the VP.S ENCORE perfusion device filled

with 3 L of Krebs-Henseleit (KH) buffer supplemented with 20PEG and hemoglobin-based oxygen carrier (HBOC; Hemoglobin Oxygen Therapeutics, LLC, Souderton, PA, USA). The heart was perfused with a pressure of approximately <20 mmHg resulting in a coronary perfusate flow of 50–100 mL/min at a 20–23°C temperature. Hearts were immediately treated with 3 mL of 3% lactic acid containing 135 mg of SNC80 (100 µM final concentration), administered through the arterial line followed by the 3 mL of KH flush. Post drug administration, the heart continued to be perfused for additional 6 hr.

Two oxygen probes (Pyro FireSting) collected oxygen partial pressure (mmHg) data of the solution before entering the coronary arteries (arterial) and after exiting the cannulated pulmonary artery (venous), whereas flow was measured using the Sensirion flow meter. Oxygen consumption was captured using the following formula, $((([O_2]a - [O_2]v)/100 * Q)/heart weight) * 100 \cdot [O_2]a$ and $[O_2]v$ are defined as oxygen content of arterial and venous perfusate respectively. $[O_2]$ was calculated as $(1.34 * Hb * SO_2) + (K * pO_2)$, where 1.34 is mL $O_2$/Hb (g), Hb is the concentration of hemoglobin measured in g/dL, $SO_2$ is the oxygen saturation ratio calculated using the equation developed by *Severinghaus, 1979*, K is the oxygen solubility coefficient adjusted for the perfusate temperature at every data point, $pO_2$ is the partial pressure of oxygen in mmHg for the perfusate sample, and Q is coronary flow in mL/min.

After preservation, hearts were placed on a Langendorff system with a perfusate mixture of 1:1 porcine blood and KH buffer. Myocardial oxygen consumption ($MVO_2$ mL$O_2$/min/100 g) was calculated during an initial 30 min of rewarming stage (before defibrillation) and after defibrillation using two oxygen probes (Pyro FireSting, placed in arterial and venous effluent). Oxygen consumption was (*Schäfer et al., 2001*) calculated using the same approach described for the VP.S ENCORE perfusion stage. The left ventricular function was expressed as dP/dT (mmHg/s) measured by placing a pressure catheter (Millar, 5F) in the LV. Data was analyzed using PowerLab (LabChart 8.1.16) blood pressure analyzer. Data was selected from the left ventricular pressure waveform as the average of 30 beats after stabilization (approx. 30 min after defibrillation). Contractility after ionotropic support was measured less than 5 min after administration of epinephrine (0.5 mg injected arterial). End diastolic pressure of the LV was maintained at 5.14±1.89 mmHg through administering perfusate directly into the LV through the pulmonary veins and venting to atmosphere.

ECG data were obtained through the placement of leads on the apex (positive), right atrial appendage (negative), and aortic root (ground). Data was collected through the PowerLab system and 30 s time intervals were selected for all groups before the flush, before defibrillation, at native contraction, and after epinephrine drug administration. The ECG amplitude (mV) was analyzed using LabChart Pro (v8.1.19) peak analysis detection at approximate timepoints 10 min (before flush) and 30 min (before defibrillation) to detect the average amplitude of electrical activity without full contraction. LabChart Pro (v8.1.19) ECG Analysis detection was used at timepoints 35 min (Epinephrine) and 60 min (Native) to detect the average amplitude of the QRS complex with contraction. The native QRS interval (s) and ST height (mV) were averaged over a 30 s time interval using LabChart Pro (v8.1.19) ECG Analysis.

After the experiment, biopsy tissue samples were collected from SNC80 and vehicle-treated samples for RNA-Seq analysis and histological evaluation. Total RNA from cardiac tissue biopsies (2.5 mm) were extracted following the RNeasy Mini kit (QIAGEN) according to the manufacturer's instructions. The concentration of total RNA was evaluated and measured at 260/280 nm by spectrophotometer (NanoVue Plus, GE Healthcare). Synthesis of cDNA was performed from 0.5 µg of total RNA, which was reverse transcribed using iScript cDNA Synthesis Kit, Bio-Rad, according to the manufacturer's instructions. Gene-specific pre-designed oligonucleotide primers were purchased from Sigma. qRT-PCR was done using SsoAdvanced Universal SYBR Green Supermix and SFX96 Touch real-time PCR detection system (Bio-Rad, T100 Thermal Cycler). The cycling parameters were as follows: initial denaturation 95°C, 2 min; denaturation 95°C, 5 s; annealing/extension 60°C, 30 s; number of cycles 40; melt curve 65–95°C (0.5°C increments). The comparative CT (2-ΔΔCT) method was used for all quantification. Values were normalized to the GAPDH housekeeping gene. Myocardial degeneration, myocardial hemorrhage, interstitial edema, and endothelial changes were semiquantitatively graded from histology images on a scale of 0–3 by a clinician blinded to sample and intervention using previously published guidelines (*Trahanas et al., 2016*).

## Porcine limb ex vivo perfusion studies

Hindlimbs (N=8) were procured via protocol tissue sharing at the Joint Base San Antonio Clinical Investigation and Research Support (CIRS) building, following IV Pentobarbital euthanasia, 100 mg/kg from a host protocol. Host protocol animals were treated with the same surgical manipulations and no chemicals or drugs were introduced experimentally prior to euthanasia. Following euthanasia, a curvilinear incision was made around the hindlimb circumferentially. Subsequently, the neurovascular pedicle in the medial hindlimb was identified and carefully dissected out to preserve the vascular bundle. The pedicle was then divided at the origin of the artery and vein to preserve maximal length of the vessels. Hindlimb bone was separated using a Gigli Saw. Hindlimbs were cannulated through the femoral artery and underwent immediate arterial flush with up to 200–400 mL heparinized-saline solution (10,000 units of heparin per 1 L of saline), until venous outflow was clear. Additionally, the femoral vein was cannulated for measurement and collection of venous outflow samples.

Limbs were divided into two groups, control (N=3) and experimental SNC80 (N=5). All hindlimbs were connected to the ENCORE head for perfusion via the cannulated femoral artery. The femoral vein was cannulated for obtaining venous samples. Hindlimbs were perfused with oxygenated KH at ambient temperature. Arterial and venous $pO_2$ were assessed hourly using a commercial blood gas analyzer (ALB1000 by Radiometer, Brea, CA, USA). Oxygen consumption was calculated from A-V differences obtained from arterial and venous blood gas readings. After 3 hr of perfusion (baseline), torpor-inducing compound SNC80 (44.96 µg/mL) was administered via the femoral artery to the experimental group (SNC80) (N=5) and an equal volume of the sham vehicle to the control limbs (N=3) followed by an additional 3 hr of perfusion (drug). The hindlimbs were then removed from the perfusor and flushed with fresh KH perfusate and replaced into the perfusor containing fresh KH solution (flush). The perfusion then resumed for an additional 6 hr. Relative metabolic rate was defined as the OCR over time, normalized to baseline measurements. The hindlimbs were weighed pre and post perfusion. Biopsy tissue samples were taken at baseline (pre-ENCORE), and before and after the administration of the SNC80 compound for RNA-Seq analysis and histological evaluation.

Biopsies for histopathology were obtained superficially from the rectus femoris muscle bundle immediately after amputation and then 24 hr after the perfusion period. Biopsies were stored in 10% formalin for 24 hr and then stored for 3 days in 70% ethanol and embedded in paraffin. The samples were sectioned into 5 µm slides and stained with H&E to evaluate tissue morphology, intracellular distance, muscle bundle area, and periodic acid-Schiff to evaluate glycogen content. Histological analysis was performed by visualizing slides under a light microscope (AM Scope T670Q, Irvine, CA, USA) and capturing images using a high-definition camera (AM Scope Model# MU2003B1, Irvine, CA, USA). From each slide, 10 digital images from randomly selected fields were obtained at ×10 magnification. The images were de-identified and evaluated for intracellular distance, muscle bundle area, and glycogen content by two independent reviewers using ImageJ software (v1.51.).

## Statistical analysis

All graphing and statistical analyses were performed with Prism 9 (GraphPad Software Inc, La Jolla, CA, USA) with a (two-sided) significance level of 0.05. Sample sizes, statistical tests, and corrections for multiple comparisons are described in each figure panel. Samples sizes were determined based on previously published reports, the capacity of each measurement system, and availability of porcine samples. All animals and samples were randomly allocated into groups and samples were not blinded.

## Software availability statement

A custom Python script was used to automatically count heart beat in *Xenopus* tadpoles. This code can be accessed on GitHub: https://github.com/k-shcherb/heart_seg, (copy archived at ***k shcherb, 2024***).

## Acknowledgements

The authors gratefully acknowledge funding from the Army Research Office/DARPA under Cooperative Agreement Number W911NF-19-2-0027. The views and conclusions contained in this document are those of the authors and should not be interpreted as representing the official policies, either expressed or implied, of the Army Research Office/DARPA or the U.S. Government. The U.S.

Government is authorized to reproduce and distribute reprints for Government purposes notwithstanding any copyright notation herein. MPO acknowledges financial support from the Margarita Salas postdoctoral grant (UNI/551/2021). We thank E Switzer for *Xenopus* embryo fertilization; R Colon, S Sundar, and E Lederer for *Xenopus* embryo husbandry and transport organization; members of the Levin Lab, Tufts University for useful discussions; and R Gould for help with database extraction. Image analysis workflows were created in consultation with D Stirling and B Cimini of the Broad Institute Imaging Platform.

## Additional information

### Competing interests

Megan M Sperry: Inventor on a relevant patentapplication held by Harvard University (PCT/US2021/012626). Erica Gardner: Inventor on a relevant patentapplication held by Harvard University (PCT/US2021/012626). Hold equity in and is employed by Unravel Biosciences, Inc. Takako Takeda: Inventor on a relevant patent807 application held by Harvard University (PCT/US2021/012626). Kristina Andrijauskaite, Exal Cisneros, Riley Lopez, Isabella Cano, Zachary Maxwell, Israel Jessop, Rafa Veraza, Leon Bunegin: Employee of Vascular Perfusion Solutions. Michael Levin: Inventor on a relevant patent application held by Harvard University (PCT/US2021/012626). Hold equity in and is a member of the scientific advisory board of Unravel Biosciences, Inc. Richard Novak: Inventor on a relevant patent application held by Harvard University (PCT/US2021/012626). Holds equity in, is a member of the board of directors, and is a current employee of Unravel Biosciences, Inc. Donald E Ingber: Founder, board member and Scientific Advisory Board chair of, and holds equity in, Emulate Inc Inventors on a relevant patent application held by Harvard University (PCT/US2021/012626). Holds equity and is a member of the board of directors of Unravel Biosciences, Inc Member of the scientific advisory board of Vascular Perfusion Solutions. The other authors declare that no competing interests exist.

### Funding

| Funder | Grant reference number | Author |
| --- | --- | --- |
| Army Research Office | W911NF-19-2-0027 | Michael Levin<br>Richard Novak<br>Donald E Ingber |
| Margarita Salas Postdoctoral grant | UNI/551/2021 | Maria Plaza-Oliver |

The funders had no role in study design, data collection and interpretation, or the decision to submit the work for publication.

### Author contributions

Megan M Sperry, Berenice Charrez, Kanoelani Pilobello, Zohreh Izadifar, Tiffany Lin, Kristina Andrijauskaite, Formal analysis, Investigation, Writing – original draft, Writing – review and editing; Haleh Fotowat, Abigail Kuelker, Sahith Kaki, Michael Lewandowski, Maria Plaza-Oliver, Exal Cisneros, Riley Lopez, Isabella Cano, Zachary Maxwell, Israel Jessop, Jorge J Pena, Diandra M Wood, Zachary T Homas, Cody J Hinshaw, Jennifer Cox-Hinshaw, Formal analysis, Investigation, Writing – review and editing; Erica Gardner, Joel Moore, Katherine Sheehan, Jaclyn Yracheta, Olivia G Parry, Justin J Sleeter, Investigation, Writing – review and editing; Shanda Lightbown, Resources, Project administration, Writing – review and editing; Ramses Martinez, Susan Marquez, Resources, Methodology, Writing – review and editing; Adama M Sesay, Supervision, Writing – review and editing; Kostyantyn Shcherbina, Resources, Software, Formal analysis, Methodology, Writing – review and editing; Takako Takeda, Daniela Del Campo, Formal analysis, Writing – review and editing; Rafa Veraza, Conceptualization, Formal analysis, Supervision, Investigation, Writing – review and editing; Leon Bunegin, Conceptualization, Formal analysis, Supervision, Writing – review and editing; Thomas J Percival, Erik K Weitzel, Michael Super, Conceptualization, Supervision, Writing – review and editing; Michael Levin, Supervision, Funding acquisition, Writing – review and editing; Richard Novak, Donald E Ingber,

Conceptualization, Supervision, Funding acquisition, Writing – original draft, Writing – review and editing

### Author ORCIDs
Megan M Sperry ⬡ https://orcid.org/0000-0001-8310-5358
Haleh Fotowat ⬡ https://orcid.org/0000-0003-0372-4912
Zohreh Izadifar ⬡ https://orcid.org/0000-0002-5926-5789
Katherine Sheehan ⬡ https://orcid.org/0009-0000-1050-7671
Richard Novak ⬡ https://orcid.org/0000-0003-0485-4284
Donald E Ingber ⬡ https://orcid.org/0000-0002-4319-6520

### Ethics

Xenopus embryos were fertilized at Tufts University using procedures reviewed and approved by the Tufts University Institutional Animal Care and Use Committee regulations (M2020-35) and transferred to the Wyss Institute. All animal experiments and procedures were reviewed and approved by the Harvard Medical School (HMS) Institutional Animal Care and Use Committee regulations (IS00000658-6). Porcine heart studies were conducted under the UT Health San Antonio approved Institutional Animal Care and Use Committee (IACUC) protocol 20210080AR. For porcine limb studies, hindlimbs were procured via protocol tissue sharing at the Joint Base San Antonio Clinical Investigation and Research Support (CIRS) building.

Reviewer #1 (Public Review): https://doi.org/10.7554/eLife.93796.3.sa1
Reviewer #2 (Public Review): https://doi.org/10.7554/eLife.93796.3.sa2
Author response https://doi.org/10.7554/eLife.93796.3.sa3

---

# Additional files

### Supplementary files

• Supplementary file 1. Clinical interpretation of electrocardiogram (ECG) in SNC80 and vehicle-treated hearts.

• Supplementary file 2. Compound synthesis description.

• MDAR checklist

### Data availability

The proteomics data reported in this paper have been deposited in the Mass Spec Interactive Virtual Environment, MassIVE, (accession no. MSV000091410). Source data for each figure and figure supplement can be accessed through Harvard University's data repository system https://doi.org/10.7910/DVN/JWLWRC. The files are labeled by the figure or table to which they correspond. A custom Python script was used to automatically count heart beat in *Xenopus* tadpoles. This code can be accessed on GitHub: https://github.com/k-shcherb/heart_seg (copy archived at *k shcherb, 2024*).

The following datasets were generated:

| Author(s) | Year | Dataset title | Dataset URL | Database and Identifier |
|---|---|---|---|---|
| Ingber DE | 2023 | GNPS - Identification of a pharmaceutical biostasis inducer that slows metabolism in multiple vertebrates that do not hibernate (Thermal Proteome Profiling in *Xenopus laevis*) | https://massive.ucsd.edu/ProteoSAFe/dataset.jsp?accession=MSV000091410 | MassIVE, MSV000091410 |
| Sperry M | 2024 | Identification of pharmacological inducers of a reversible hypometabolic state for whole organ preservation | https://doi.org/10.7910/DVN/JWLWRC | Harvard Dataverse, 10.7910/DVN/JWLWRC |

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
