## [Editor Report · eLife assessment]

Pharmacological induction of physiological slowing combined with organ perfusion systems could provide a novel therapeutic strategy for tissue and organ preservation. Using a *Xenopus* model, the authors provide **important** findings on a use of drug to slow down metabolism for the purpose of organ preservation. The authors provide **compelling** evidence that SNC80 can rapidly and reversibly slow biochemical and metabolic activities while preserving cell and tissue viability. This approach may be beneficial for transplantation, trauma management, and improving organ survival in remote and low-resource settings

---

## [Referee Report · Reviewer #1 (Public Review)]

Summary:

In this study of metabolism using *Xenopus*, explanted porcine hearts and limbs, and human organs-on-chips, Sperry et al studied the ability of WB3 to slow metabolism and mobility. The group developed WB3, an analog of SNC80, void of SNC80's delta-opioid receptor binding capacity and studied its metabolic impact. The authors concluded that SNC80 and its analog WB3 can induce "biostasis" and produce a hypometabolic state which holds promise for prolonging organ viability in transplant surgery as well as other potential clinical benefits.

Strengths:

This study also opens new avenues for therapeutic possibilities in areas such as trauma, acute infection, and brain injuries. The overall methodology is acceptable, but certain concerns should be addressed.

Weaknesses:

Major comments:

(1) In cardiac and renal transplantation, cold preservation in ice remains a common practice for transporting explanted hearts to donors which remains a cheap and easily accessible way of preserving organs. While ex-vivo mechanical circulatory platforms have been developed and are increasingly being utilized to prolong organ viability, cold preservation remains widely used. The authors perfused explanted hearts with oxygenated perfusion preservation devices at subnormothermic temperatures (20-23C) which is even much lower than routinely used in clinical cardiopulmonary bypass scenarios (28-32C) (in the discussion, the authors allude to SNC80's possible "protective effect" in cardiac bypass). It is unclear how much of the hypometabolic state is related to WB3 administration versus hypothermia. The study will benefit from a comparison of WB3 administration and hypothermia in *Xenopus*, explanted porcine organs versus cold preservation alone to show distinction in biostasis parameters.

(2) The authors selected SNC80 based on a literature survey where it was identified based on its ability to induce hypothermia and protect against the effects of spinal cord ischemia in rodents. While this makes sense, were other drugs (eg. Puerarin) considered? The induction of hypothermia and spinal cord protective effect of SNC80 may be multifactorial and not necessarily related to its biostatic effects as the authors describe. Please provide some more context into the background of SNC80.

(3) In most of the models, the primary metric that the authors utilize to characterize metabolic activity is oxygen consumption, which is a somewhat limited indicator. For instance, this does not provide any information, however, on anaerobic metabolic activity. In addition, the ATP/ADP ratio was found to decrease in the organ chips where SNC80 was utilized, but similar findings were not presented for the other models.

(4) The authors should provide a more detailed explanation of SNC80's mechanisms of interaction with proteins related to transmembrane transport, mitochondrial activity, and metabolic processes. What is the impact of SNC80 on mitochondrial function, particularly ATP production and mitochondrial respiration? Are there changes in mitochondrial membrane potential, electron transport chain activity, or oxidative phosphorylation? In this context, authors discuss the potential role of NCX1 as a binding target for SNC80 and its various mechanisms in slowing metabolism. However, no experiments have been done to confirm this binding in the present study. Co-immunoprecipitation studies using appropriate antibodies against SNC80 and NCX1 should be considered to demonstrate their direct binding. Additionally, surface plasmon resonance (SPR) or isothermal titration calorimetry (ITC) experiments could be employed to quantify the binding affinity between SNC80 and NCX1, providing further evidence of their interaction. These experiments would elucidate the binding mechanism between SNC80 and NCX1 and reveal more information on the mechanism of action for SNC80.

(5) The manuscript notes that histological analysis was conducted, but it seems that only example images are provided, such as Fig 4f. Quantified histological data would provide a more thorough understanding of tissue integrity.

(6) Some of the points mentioned in the discussion and conclusion are rather strong and based on possible associations such as SNC80's potential vasodilatory capacity conferring a cardioprotective effect, ability to reversibly suppress metabolism across different temperatures and species. Please tone this down and stay limited to the organs studied. Further, the reversibility of the findings may be more objectively assessed by biomarkers with decreased immunofluorescence in response to ischemia such as troponin I for heart and albumin for liver. Additionally, an investigation of proteins involved in inflammation, hypoxia, and key cell death pathways using immunohistochemistry analysis can better describe the impact of treatment on apoptosis/necroptosis.

(7) What could be the underlying cause of the observed increase in intercellular spacing after SNC80 administration in porcine limbs which also seems to be evident in the heart histology samples? This seems to be more prominent in the SNC80 compared to the vehicle group.

(8) In the Discussion section, it would be valuable to provide a concise interpretation of the lipidomic data, particularly explaining how changes in acylcarnitine and cholesterol ester levels may relate to tadpole metabolism, hibernation, or other biological processes.

(9) What are the limitations or disadvantages of the study? Does SNC80 possess any immunomodulatory properties that might affect the outcomes of organ transplantation? Are there specific organs for which SNC80 may not be a suitable preservation agent, and if so, what are the reasons behind this?

Comments on revised version:

The authors have satisfactorily addressed our comments in the rebuttal letter. The limitations described by the authors in point #9, however, need to be incorporated in the revised manuscript in detail as they are important in guiding interpretation of the present data. Congratulations again on the important study.

---

## [Referee Report · Reviewer #2 (Public Review)]

Summary:

This manuscript titled "Identification of pharmacological inducers of a reversible hypometabolic state for whole organ preservation" reports the effects of delta opioid receptor activator SNC80 and its modified analog WB3 with ~1,000 times less delta opioid receptor binding activity on metabolic state.

Strengths:

This is an interesting study with potentially broad implications for organ preservation.

Weaknesses:

However, there are several limitations which raise concerns.

(1) The authors developed an analog of a known delta opioid receptor activator SNC80 with three orders of magnitude lesser binding with the delta opioid receptor WB3. This will likely reduce the undesirable effects of SNC80 while preserving metabolic slowing needed for organ preservation. Yet, most experiments were done with SNC80, not the superior modification, WB3, shown in only a limited set of experiments, Figure 3.

(2) The heart is one of the most challenging organs to preserve, and some experiments are done to establish the metabolic effects of SNC80. However, the biodistribution study, shown in Figure 2, conspicuously omitted the heart.

(3) I do not understand the design of the electrophysiology and contractility experiments with the porcine hearts. How did you defibrillate the hearts after removal and establishing perfusion? Lines 173-175 on Page 7 state: "After defibrillation with epinephrine, the P and QRS waveforms were visible in ECGs from 3 of 4 SNC80-treated hearts (Table S1), suggesting that those hearts regain atrial and ventricular polarization." Please clarify. Defibrillation is done with an electric shock. Also, please show the ECG recordings to support your conclusions about "polarization." What did you mean by "polarization"? Depolarization? Repolarization? Or resting potential. To establish a normal physiological state, please show ECG waveforms and present data on basic ECG characteristics: heart rate, PQ and QT intervals, and P and QRS durations. I recommend perfusion of the porcine heart with WB3, not only SNC80.

(4) Pathology data also raises concerns. The histology images shown in Figure 4f are not quantified, and they show apparently higher levels of tissue disruption in SNC80-treated tissue vs vehicle-treated. The test (lines 169-171) confirms this concern: "In some hearts treated with SNC80, greater waviness of muscle fibers was observed, possibly indicating a state of muscle contraction." It will be helpful to measure markers of apoptosis and necrosis and to apply TTC viability staining.

(5) The apparent state of contracture suggests a higher degree of myocardial damage and a high intracellular calcium level in SNC80-treated hearts. The authors suggested that the sodium-calcium exchanger NCX is a possible target of SNC80 and could be responsible for the "hypometabolic state." However, NCX1 is critically important in the extrusion of cytosolic Ca2+ during the diastolic phase. Failure to remove excessive calcium and restore ionic homeostasis would lead to calcium overload and heart failure.

(6) I am surprised the authors did not consider using the gold standard assay for measuring mitochondrial function in cells by the Seahorse Cell Mito Stress Test.

Comments on revised version:

I am satisfied with the revisions. The authors addressed major concerns with new data and/or provided satisfactory rebuttal.

---

## [Author Response]

The following is the authors’ response to the original reviews.

**Public Reviews:**

**Reviewer #1:**
(1) In cardiac and renal transplantation, cold preservation in ice remains a common practice for transporting explanted hearts to donors which remains a cheap and easily accessible way of preserving organs. While ex-vivo mechanical circulatory platforms have been developed and are increasingly being utilized to prolong organ viability, cold preservation remains widely used. The authors perfused explanted hearts with oxygenated perfusion preservation devices at subnormothermic temperatures (20-23C) which is even much lower than routinely used in clinical cardiopulmonary bypass scenarios (28-32C) (in the discussion, the authors allude to SNC80's possible "protective effect" in cardiac bypass). It is unclear how much of the hypometabolic state is related to WB3 administration versus hypothermia. The study will benefit from a comparison of WB3 administration and hypothermia in *Xenopus*, explanted porcine organs versus cold preservation alone to show distinction in biostasis parameters.

Indeed, we expect that both pharmaceutical interventions and cooling could contribute to a hypometabolic state. To assess this, the controls and the treated groups were exposed to the same temperatures for both the *Xenopus* (18C) and porcine heart experiments (20-23C). Therefore, we can conclude that any changes in the treatment group relative to control can be attributed to the introduction of SNC80 or WB3 and not from cooling alone.

(2) The authors selected SNC80 based on a literature survey where it was identified based on its ability to induce hypothermia and protect against the effects of spinal cord ischemia in rodents. While this makes sense, were other drugs (eg. Puerarin) considered? The induction of hypothermia and spinal cord protective effect of SNC80 may be multifactorial and not necessarily related to its biostatic effects as the authors describe. Please provide some more context into the background of SNC80.

During our research program, we considered and tested other drugs (>100 existing compounds in *Xenopus* screens). Although the published hypothermic and tissue protective effects suggested to us that SNC80 should be included in screening, it was not until we observed effects across multiple test parameters, systems, and species that we honed in on SNC80 as a lead compound. We have added additional information to further clarify the background of SNC80 on pgs. 3-4.

(3) In most of the models, the primary metric that the authors utilize to characterize metabolic activity is oxygen consumption, which is a somewhat limited indicator. For instance, this does not provide any information, however, on anaerobic metabolic activity. In addition, the ATP/ADP ratio was found to decrease in the organ chips where SNC80 was utilized, but similar findings were not presented for the other models.

We thank reviewers for their important point. We have therefore added additional experiments, including the Seahorse Mitostress assay for the four human cell types (Caco-2, Huh7, LSEC and HUVEC) used in the Organ Chip systems. We have added a description and an interpretation of the results in the section, Stasis induction in cultured human cells and tissues and mention the role of glycolysis and cytosolic reductive carboxylation as compensatory mechanisms. Although the ATP/ADP ratio gave us useful insight into Huh-7 cells and chips metabolic activity, this method requires transfection and live imaging which does not suit other models such as *Xenopus*, or whole organs. Additionally, in animal models there may be other confounding factors that might influence ATP/ADP.

(4) The authors should provide a more detailed explanation of SNC80's mechanisms of interaction with proteins related to transmembrane transport, mitochondrial activity, and metabolic processes. What is the impact of SNC80 on mitochondrial function, particularly ATP production and mitochondrial respiration? Are there changes in mitochondrial membrane potential, electron transport chain activity, or oxidative phosphorylation? In this context, the authors discuss the potential role of NCX1 as a binding target for SNC80 and its various mechanisms in slowing metabolism. However, no experiments have been done to confirm this binding in the present study. Coimmunoprecipitation studies using appropriate antibodies against SNC80 and NCX1 should be considered to demonstrate their direct binding. Additionally, surface plasmon resonance (SPR) or isothermal titration calorimetry (ITC) experiments could be employed to quantify the binding affinity between SNC80 and NCX1, providing further evidence of their interaction. These experiments would elucidate the binding mechanism between SNC80 and NCX1 and reveal more information on the mechanism of action for SNC80.

We agree that further definition of the mechanism of action is an important next step for this work; however, it is far beyond the scope of the present study.

(5) The manuscript notes that histological analysis was conducted, but it seems that only example images are provided, such as Figure 4f. Quantified histological data would provide a more thorough understanding of tissue integrity.

We have added quantified histological data to the manuscript that was performed by a clinician blinded to the groups and interventions (Figure 4f).

(6) Some of the points mentioned in the discussion and conclusion are rather strong and based on possible associations such as SNC80's potential vasodilatory capacity conferring a cardioprotective effect, and ability to reversibly suppress metabolism across different temperatures and species. Please tone this down and stay limited to the organs studied. Further, the reversibility of the findings may be more objectively assessed by biomarkers with decreased immunofluorescence in response to ischemia such as troponin I for the heart and albumin for the liver. Additionally, an investigation of proteins involved in inflammation, hypoxia, and key cell death pathways using immunohistochemistry analysis can better describe the impact of treatment on apoptosis/necroptosis.

We have revised aspects of the Discussion and Conclusion to focus on the organs studied in the present work (pgs. 14-17). We agree that markers of inflammation, hypoxia, and cell death are critical for assessing tissue health post-treatment. We performed PCR to assess such markers (Figure 4e) and found reductions in inflammatory cytokine and injury biomarker levels. Although we agree that immunohistochemistry may be useful, such as for looking at any spatial patterns of injury, PCR offers broader dynamic range and higher sensitivity and therefore was chosen for this assay.

(7) What could be the underlying cause of the observed increase in intercellular spacing after SNC80 administration in porcine limbs which also seems to be evident in the heart histology samples? This seems to be more prominent in the SNC80 compared to the vehicle group.

Since the muscle bundle areas of baseline and treated tissues were essentially the same, the increase in intracellular space in the SNC80-treated tissue suggests a compensatory reduction in muscle fiber diameter. Intracellular metabolite concentrations have been shown to be quite stable over a large range of metabolic activities (Hochachka et al. 1998). As such, a reduction in metabolic activity induced by SNC80 may suggest reduction in the accumulation of intracellular metabolites. In order to maintain a stable intracellular metabolite concentration, water would have to be expelled accounting for the increased intracellular space.

P W Hochachka, G B McClelland, G P Burness, J F Staples, R K Suarez Comp Biochem Physiol B Biochem Mol Biol 120, 17–26 (1998).

(8) In the Discussion section, it would be valuable to provide a concise interpretation of the lipidomic data, particularly explaining how changes in acylcarnitine and cholesterol ester levels may relate to tadpole metabolism, hibernation, or other biological processes.

An interpretation of the lipidomics data has been summarized in the Discussion (pg. 14).

(9) What are the limitations or disadvantages of the study? Does SNC80 possess any immunomodulatory properties that might affect the outcomes of organ transplantation? Are there specific organs for which SNC80 may not be a suitable preservation agent, and if so, what are the reasons behind this?

This study is limited in two ways. The first is that we characterized the function of the donor pig heart outside of the body, and therefore future work will be required to verify the function and quality of the hearts after they have been transplanted. Secondly, SNC80 is not currently approved for use in clinical settings and during earlier pre-clinical trials of the drug, side effects including seizures were noted and its development was halted. It is hypothesized that these seizures are related to SNC80’s delta opioid activity, so we developed a new, non-opioid analog called WB3, which will be used in future work. We have added a description of the prior seizure findings to the text (pg. 5).

Based on assessment of tissue biomarkers by PCR, it seems that SNC80 does exhibit immunomodulating properties. Because organ transplant recipients are treated with strong immunosuppressants to prevent organ rejection, we anticipate that SNC80 would either further support this goal, have little additional effect, or reduce the amount of additional immunosuppressive drugs that would need to be administered. To date, our data does not suggest that there are specific organs for which SNC80 may not be a suitable preservation agent.

**Reviewer #2:**
(1) The authors developed an analog of a known delta opioid receptor activator SNC80 with three orders of magnitude lesser binding with the delta opioid receptor WB3. This will likely reduce the undesirable effects of SNC80 while preserving the metabolic slowing needed for organ preservation. Yet, most experiments were done with SNC80, not the superior modification, WB3, shown in only a limited set of experiments, Figure 3.

We included the WB3 studies in *Xenopus* to confirm that the biostatic activity is not mediated through the delta opioid receptor. We have only performed a limited number of experiments with WB3 because we are focused on improving its solubility so that it can be easily dissolved in common organ perfusates without DMSO, which we were able to use in the *Xenopus* experiments.

(2) The heart is one of the most challenging organs to preserve, and some experiments are done to establish the metabolic effects of SNC80. However, the biodistribution study, shown in Figure 2, conspicuously omitted the heart.

Thank you for this suggestion. We returned to the biodistribution study dataset and were able to measure uptake by the heart at the 1-hour time point. We observe an increase in uptake above levels observed for other tissues at 1 hour and at levels similar to the skeletal muscle at 2 hours (plot below). Unfortunately, the heart was not visible in a sufficient number of *Xenopus* tissue sections to reevaluate uptake at the 2-hour time point. We were also able to re-evaluate the lipidomics data for the heart. Acylcarnitine and cholesterol ester were not significantly different between vehicle and SNC80-treated groups. The lack of change in acylcarnitine is particularly important since its upregulation has been shown to be a marker for cardiovascular disease in humans (Deda et al. 2022). The expanded lipidomics data have been added to Figure 2.

Deda O, Panteris E, Meikopoulos T, Begou O, Mouskeftara T, Karagiannidis E, Papazoglou AS, Sianos G, Theodoridis G, Gika H. Correlation of serum acylcarnitines with clinical presentation and severity of coronary artery disease. Biomolecules. 2022 Feb 23;12(3):354.

(3) I do not understand the design of the electrophysiology and contractility experiments with the porcine hearts. How did you defibrillate the hearts after removal and establishing perfusion? Lines 173-175 on Page 7 state: "After defibrillation with epinephrine, the P and QRS waveforms were visible in ECGs from 3 of 4 SNC80-treated hearts (Table S1), suggesting that those hearts regain atrial and ventricular polarization." Please clarify.Defibrillation is done with an electric shock. Also, please show the ECG recordings to support your conclusions about "polarization." What did you mean by "polarization"? Depolarization? Repolarization? Or resting potential. To establish a normal physiological state, please show ECG waveforms and present data on basic ECG characteristics: heart rate, PQ and QT intervals, and P and QRS durations. I recommend perfusion of the porcine heart with WB3, not only SNC80.

Hearts were defibrillated by the application of a 10 to 30 Joule electrical shock delivered from internal paddles positioned at the right atrium (negative) across to the left ventricle (positive). Once rhythm was established, 0.5 ml of 1:1000 epinephrine was administered via the aortic inflow. Electrocardiogram (ECG) showed that both vehicle and SNC80-treated hearts exhibited irregular contractions after perfusate flush and during rewarming prior to defibrillation. After defibrillation (10-30 J electrical shock) followed by epinephrine, a regular heartbeat was established in 3 of 4 SNC80-treated hearts, exhibiting normal P and QRS waveforms (Table S1). That observation suggested that the intrinsic atrial and ventricular muscle fiber contractility was preserved, and the overall conduction system of the heart was viable. The pulse rates of SNC80-treated hearts were at or near normal for porcine hearts (70-120 beats/min) after defibrillation. Vehicle-treated hearts exhibited tachycardia following defibrillation, with all exhibiting pulse rates above the normal range for porcine hearts. We have added clarifying text and definitions (pg. 8). We have only performed a limited number of experiments with WB3 because we are focused on improving its solubility so that it can be easily dissolved in common organ perfusates without DMSO, which we were able to use in the *Xenopus* experiments.

(4) Pathology data also raises concerns. The histology images shown in Figure 4f are not quantified, and they show apparently higher levels of tissue disruption in SNC80-treated tissue vs vehicle-treated. The test (lines 169-171) confirms this concern: "In some hearts treated with SNC80, greater waviness of muscle fibers was observed, possibly indicating a state of muscle contraction."

The histology images shown in Figure 4f were quantified and the myocardial injury score quantification show comparable histology between the groups.

(5) The apparent state of contracture suggests a higher degree of myocardial damage and a high intracellular calcium level in SNC80-treated hearts.The authors suggested that the sodium-calcium exchanger NCX is a possible target of SNC80 and could be responsible for the "hypometabolic state." However, NCX1 is critically important in the extrusion of cytosolic Ca2+ during the diastolic phase. Failure to remove excessive calcium and restore ionic homeostasis would lead to calcium overload and heart failure.

The histological assessment doesn’t indicate a higher degree of myocardial damage in SNC80 treated hearts. Our data are not suggestive of high intracellular calcium buildup in SNC80treated hearts. If that were the case, we would have had challenges restoring the rhythm of the hearts on the Langendorff post-preservation, which was not observed.

(6) I am surprised the authors did not consider using the gold standard assay for measuring mitochondrial function in cells by the Seahorse Cell Mito Stress Test.

Thank you for this important point. We have added data from the Seahorse Mitostress assay for the four human cell types (Caco-2, Huh7, LSEC and HUVEC) included in the Organ Chip experiments. We have added a description and an interpretation of the results in the section Stasis induction in cultured human cells and tissues. We now mention the role of glycolysis and cytosolic reductive carboxylation as compensatory mechanisms.

**Reviewer #3:**
(1) The authors perform a literature search to identify SNC80 as a promising hit. However, the details of the literature search, a list of other potential hits, and the criteria for identification of SNC80 are not described. The hypometabolic effect of SNC80 exposure is well-characterized in the *Xenopus* model. Furthermore, the authors show that SNC80 localises to the brain, but do not discuss several studies that have pointed to convulsions induced by exposure to high doses of SCN80, and whether this would be apparent in the *Xenopus* studies. The authors have promising data on the WB3 morpholino that retains or even improves on the hypometabolism phenotype of SCN80 while likely not retaining delta opioid activity. However, this is not functionally demonstrated. Moreover, WB3 is not used in any of the other assays and models used in the study. In the setting of cardiac transplant surgery, co-administration of SNC80 reduces metabolic activity and inflammation, although it is unclear if there is an improvement in recovery of organ function due to SCN80.

Thank you for raising these important points. We have added details of the process to identify SNC80 (pgs. 3-4) and a discussion of the studies pointing to convulsions with high doses of SNC80 (pg. 5) (which were not observed in *Xenopus* studies). We have also incorporated measurements of oxygen consumption during WB3 treatment in *Xenopus* (Figure 3d).

(2) The reversible induction of hypometabolic status is also demonstrated in two different organ chips. These models could identify the differential response of epithelial cells and vascular cells to drug perfusion, but the authors have mostly focused on the former. Finally, the authors identify specific targets for the hypometabolic effect of SNC80, which is a valuable resource for other screening studies and can form the basis for future work.

In the revised manuscript, we have also added data from the Seahorse Mitostress assay for the four human cell types (Caco-2, Huh7, LSEC and HUVEC). We have added a description and an interpretation of the results in the section Stasis induction in cultured human cells and tissues. We highlight the differences in metabolic response from the four cell types to SNC80 treatment. It is important to note that the metabolism-suppressing effects of SNC80 were most potent in the epithelial cells that were originally derived from highly metabolic tumors (Caco-2 and Huh7) versus primary normal endothelial cells (HUVEC and LSEC), which is also consistent with past work suggesting that targeting of the NCX1 channel might offer a way to slow tumor growth (Wan et al. 2022). Because we observed more prominent effects in epithelial cells in 2D assays, we decided to focus the 3D organ chips assays on epithelial cells.

Wan, H. et al. NCX1 coupled with TRPC1 to promote gastric cancer via Ca2+/AKT/β-catenin pathway. Oncogene (2022) doi:10.1038/s41388-022-02412-9.

**Recommendations for the authors:**

**Reviewer #1:**
(1) Line 136, "Based on these intriguing findings with human Organ Chips". No mention of human organ chips was made in the text at this point, suggest rewording.

Thank you for identifying this error. We have revised this line (pg. 6).

(2) Please provide more information on previous studies that have explored other drugs for organ protection, the novelty of the findings of this study, and how the findings of this study compare to prior data.

Building on the background of organ preservation drugs provided in the Introduction, we have added details to compare our outcomes to other drugs explored for organ protection (pg. 15).

(3) The dosing study in Supplemental Figure S1 provides some context on why the authors utilized the 100 uM SNC80 concentration. It would be helpful if the authors could elaborate in the Discussion on the mechanistic rationale for this concentration.

This dose was chosen to maximize suppression of metabolic and activity parameters, while ensuring reversibility of biostasis. We have clarified this in the Discussion (pg. 14).

(4) In Supplement Figure S2a, the y-axis measures the relative metabolic rate. It seems from the text that this is a relative measure of oxygen consumption, but it should be clarified accordingly.

We have clarified this point in the Methods section.

(5) What is the specific time or time frame when the reversed effect of SNC80 is most pronounced or at its peak?

When *Xenopus* are moved to fresh medium after SNC80 treatment, we observe a 15-minute period during which no reversal is evident from motion measurements. After that period, we observe a gradual, linear recovery over 2 hours. We cannot designate a specific period in which the reversal effect is most pronounced from these data.

(6) WB3 seems to show a faster and stronger impact on swimming in comparison to SNC80. What could be the potential reasons for this difference, and could this have any clinical implications?

From our current data, we understand the key difference to be that SNC80 has greater affinity for the delta opioid receptor compared to WB3. Therefore, we hypothesize that by not interacting with the opioid system, WB3 induces faster and stronger impacts on swimming. In mice, it has been shown that SNC80 directly inhibits forebrain GABAergic neurons via activity at their delta opioid receptors, which leads to convulsions (Chung et al. 2015). Although we do not observe seizure-like behavior in *Xenopus*, drugs that inhibit GABAergic neurons can produce stimulant effects in vivo. Since WB3 has a lower affinity for the delta opioid receptor, it likely produces less stimulation, leading to faster and stronger suppression of swimming behaviors. Additionally, it is possible that WB3 interacts with additional targets we have not yet identified.

Chung PC, Boehrer A, Stephan A, Matifas A, Scherrer G, Darcq E, Befort K, Kieffer BL. Delta opioid receptors expressed in forebrain GABAergic neurons are responsible for SNC80-induced seizures. Behavioural brain research. 2015 Feb 1;278:429-34.

(7) Elaborate on the potential significance of SNC80's distribution in the GI tract, gill region, and skeletal muscle. How might this distribution relate to the observed physiological effects?

In *Xenopus* tadpoles, we observe SNC80 uptake in the gill region and GI tract within 1 hour. The multiple possible routes of uptake in *Xenopus* (skin, gills, and mouth) may account for the relatively rapid physiological effects observed in our experiments. The uptake observed in the muscle may be specifically responsible for the slowed motion observed in *Xenopus* activity assays. This has been elaborated upon in the text (pg. 5).

(8) Please use italics where needed, e.g., in vitro, in vivo, etc.

This has been updated throughout the article.

(9) Supplemental Figure S1 - Is there any reason for having 3 replicates for the 100uM compared to the 4 replicates in the other groups?

Each group had 4 replicates; however, a review of the replicates for the 100 µM group suggested the presence of a leak or air bubble in one oxygen measurement vial, which, therefore, had to be excluded from the analysis.

(10) Figure 3 description - 'c' should be bold.

Figure 3 has been updated.

**Reviewer #3:**
Title: The title suggests that several candidate compounds are identified but the study focuses primarily on SCN80. Please consider rephrasing to make it more specific to this molecule. Alternatively, the manuscript would be significantly strengthened if more data is provided for WB3.

Although the study focuses on SNC80, we introduce an entirely novel molecule, WB3, and therefore, we feel it is more appropriate to indicate that multiple molecules were studied.

Line 58-59: please cite additional primary literature papers for the different therapeutics discussed. As an example, the authors do not cite or discuss Massen et al PMID: 31743376 which suggests that H2S is able to induce similar hypometabolic effects even at 37C.

Thank you for this suggestion. We have expanded our discussion of primary literature paper for the therapeutics discussed (pg. 15).

Line 76 - 77: The authors do not provide any data on the other possible hits from their literature search or methods details on how this was done. No relevant literature has been cited. What criteria were used to finalise SNC80?

During our research program, we considered and tested other drugs (>100 existing compounds in *Xenopus* screens). Although the published hypothermic and tissue-protective effects suggested that SNC80 should be included in screening, it was not until we observed effects across multiple test parameters, systems, and species that we honed in on SNC80 as a lead compound. We have added additional information to further clarify the background of SNC80 on pgs. 3-4.

Line 85 and Lines 342-345 in the Discussion: SNC80 is reported to induce convulsions at high doses in rodents and primates - was this also evident in the *Xenopus* studies? How does the dose used in the *Xenopus* studies compare with the high dose (ca. 10 mg/kg) used in primate studies Danielson et al., PMID: 17112570?

We did not observe convulsions in SNC80-treated *Xenopus*. However, we have updated the manuscript to include previous observations of convulsions in rodents and primates treated with SNC80 (pg. 5). Due to a number of differences, it is challenging to directly compare the dosing in *Xenopus* studies to those in the primate. In the present study, groups of 10 *Xenopus* were exposed to a 10 mL pool of 100 µM SNC80, which may be absorbed via oral, gill, and skin routes. Primates were dosed with 10 mg/kg delivered intramuscularly. Because these models may result in different drug biodistributions, any direct comparisons would be speculative. Further work in rodent models may help clarify the relevant dosing differences.

Line 117: what does 'double the concentration' mean? Is this with reference to the dose of SNC80? If so, is this sufficient to completely block opioid receptor activity?

Yes, we meant that naltrindole was dosed at double the concentration of SNC80. We have clarified this in the text (pg. 5). Prior work in rodent brain tissue has shown that radiolabeled naltrindole binds to saturation at picomolar to nanomolar concentrations (Yamamura et al. 1992). To confirm our initial observations with naltrindole and SNC80, we also tested a SNC80 analog (WB3) with very low delta opioid activity (Figure 3), which showed similar effects.

Yamamura MS, Horvath R, Toth G, Otvos F, Malatynska E, Knapp RJ, Porreca F, Hruby VJ, Yamamura HI.

Characterization of [3H] naltrindole binding to delta opioid receptors in rat brain. Life sciences. 1992 Jan 1;50(16):PL119-24.

Figure 3c, d: It appears that WB3 is even more effective at rapidly reducing motion and inducing faster recovery which is an exciting result. However, in 3d it appears that longterm exposure of 8h has detrimental effects since the heart rate remains depressed. Please clarify.

Yes, at 8 hours, we observe slow recovery and, in some cases, maintenance of depressed heart rates. This could be because the drug is more lipophilic and might remain in fat tissue for longer times. Although our current goal is to lengthen the time window for heart transplant surgery to 6 hours, we are working on formulating WB3 to optimize safety for longer applications (8+ hours).

Figure 4: the experiments with the heart transplants are well done, but do not demonstrate an additional protective effect over the current standard of care except for the reduced metabolism. Could the authors discuss this further in the discussion or provide data with WB83, which may show a stronger effect? Scale bars are missing in panel f.

In addition to reduced metabolism, we also demonstrate reduced expression of inflammation, hypoxia, and cell death-related markers compared to machine perfusion alone (Figure 4e). The potential protective effect of the biostasis-inducing compounds will be further investigated in a planned orthotopic porcine transplant study where pigs will be followed up for 6 hours post weaning off a bypass machine allowing enough time to assess potential benefit of biostasisinducing drugs. Additionally, we have added scale bars (Figure 4f).

Order of manuscript: Line 136 already refers to the organ-chip data, which is only presented at the end. Please edit. I feel the manuscript would flow better with the organchip data presented before the heart transplant data.Organ-chip data: this is an important component of the story but is only shown in supplementary figures. Consider showing this data in the main figures, as eLife has no space restrictions. Furthermore, it is unclear if the effluent collected and analysed is from apical or vascular, or both. In any case, the analysis via microscopy-based methods appears restricted to the epithelium. The manuscript would be significantly strengthened by providing some data on the effect of SNC80 on vascular cells.

As requested, we have moved the Organ Chips results to a main figure (new Fig. 5). We have added additional experiments, including the Seahorse Mitostress assay for the four human cell types (Caco-2, Huh7, LSEC and HUVEC). We have added a description and an interpretation of the results in the section Stasis induction in cultured human cells and tissues. The 2D assays showed that metabolism-suppressing effects of SNC80 were most potent in the epithelial cells that were originally derived from highly metabolic tumors (Caco-2 and Huh7) versus endothelial cells (HUVEC and LSEC). Based on these results, we decided to focus the 3D organ chips assays on epithelial cells only, and hence only analyzed effluents from the epithelial (apical) channel.

Methods section for fabrication of oxygen sensors: Please refer to prior papers from your lab (Grant et al., PMID: 35274118) with regards to details of the fabrication of the devices with inbuilt oxygen sensors.

The methods used for the fabrication of oxygen sensors will be included in a separate manuscript currently in preparation.

Figure S3 and Line 243-244: Please provide the data for untreated control organ chips in panels d and e a mean value for which is quoted in the main text. The images in panel f are too small for the reader to appreciate the point, please provide zooms. Scalebars are also missing from these images. Please increase the number of replicates for S3f - the liver-chip data has only two replicates which has very low power for statistical testing. In general, the number of organ chips used for the data for each panel is missing.

As mentioned in the captions, Figure S3 (now Figure S5) panels d and e show average albumin production of Liver Chips at day 7-10 of culture. These measurements were performed before any treatment with SNC80 to characterize the chip’s functional metabolism. In panel g, although we only show biological N=2-3, each datapoint corresponds to an average of multiple fields of view (multiple technical replicates). We have now clarified this in the figure legend.

Figure S4 - I do not quite understand why the perfusion with the vehicle only also affects oxygen release in the liver chip. Is it possible to use a different vehicle?

The liver and gut oxygen levels are not on the same y-axis (gut on the left and liver on the right). The oxygen fold change of the liver control chip is below 0.5, which is in the same range as the gut control chip (0 +/- 0.25). There is a natural variation in oxygen consumption over the lifetime of the chips (now Figure 5c), and untreated cells are metabolically active and consuming oxygen. The small drop observed suggests that liver chips may not have reached a stable oxygen consumption rate at the time of the experiment, whereas the gut chips have stabilized.

Figure S5c-f: The units on the Y-axis are missing.

Panels S5c-d (now Figure S6c-d) depict the percent cytotoxicity and are thus unitless. Panels S5e-h (now Figure S6e-h) show the effluent levels relative to baseline and are also unitless. We have updated the figure caption to clarify this.